# YAP/TAZ enhances P-body formation to promote tumorigenesis

Xia Shen[1,2†], Xiang Peng[1,2†], YueGui Guo[1,2†], Zhujiang Dai[1,2], Long Cui[1,2], Wei Yu[3], Yun Liu[1,2]*, Chen-Ying Liu[1,2]*

[1]Department of Colorectal and Anal Surgery, Xinhua Hospital, Shanghai Jiao Tong University School of Medicine, Shanghai, China; [2]Shanghai Colorectal Cancer Research Center, Shanghai, China; [3]State Key Laboratory of Genetic Engineering, School of Life Sciences, Zhongshan Hospital, Fudan University, Shanghai, China

**Abstract** The role of processing bodies (P-bodies) in tumorigenesis and tumor progression is not well understood. Here, we showed that the oncogenes YAP/TAZ promote P-body formation in a series of cancer cell lines. Mechanistically, both transcriptional activation of the P-body-related genes *SAMD4A, AJUBA,* and *WTIP* and transcriptional suppression of the tumor suppressor gene *PNRC1* are involved in enhancing the effects of YAP/TAZ on P-body formation in colorectal cancer (CRC) cells. By reexpression of PNRC1 or knockdown of P-body core genes (*DDX6, DCP1A,* and *LSM14A*), we determined that disruption of P-bodies attenuates cell proliferation, cell migration, and tumor growth induced by overexpression of YAP[5SA] in CRC. Analysis of a pancancer CRISPR screen database (DepMap) revealed co-dependencies between YAP/TEAD and the P-body core genes and correlations between the mRNA levels of *SAMD4A, AJUBA, WTIP, PNRC1,* and YAP target genes. Our study suggests that the P-body is a new downstream effector of YAP/TAZ, which implies that reexpression of PNRC1 or disruption of P-bodies is a potential therapeutic strategy for tumors with active YAP.

*For correspondence:
liuyun@xinhuamed.com.cn (YL);
liuchenying@xinhuamed.com.
cn (C-YL)

†These authors contributed
equally to this work

Competing interest: The authors
declare that no competing
interests exist.

Reviewing Editor: Yongliang
Yang, Dalian University of
Technology, China

## eLife assessment

This **valuable** study advances our understanding that YAP/TAZ, as well as their target genes, plays a prominent role in the formation of processing bodies (P-bodies). The evidence supporting the conclusions is **convincing**. The article could be improved through further analysis to elucidate the mechanistic link between P-body formation and oncogenesis. The work will be of broad interest to scientists working in the field of Hippo signaling and cancer biology.

## Introduction

The Hippo pathway is an evolutionarily conserved signaling pathway that regulates organ size and plays vital roles in development and tissue homeostasis (*Driskill and Pan, 2021*; *Ma et al., 2019*; *Russell and Camargo, 2022*). The transcriptional output of the Hippo pathway is mainly mediated by the YAP/TAZ-TEAD transcription complex. In response to various extracellular or intracellular signals, including cell–cell contact, mechanical force, serum stimulation, cellular stress, and cellular energy status, the YAP/TAZ-TEAD complex modulates target gene expression to respond to environmental cues (*Calvo et al., 2013*; *Misra and Irvine, 2018*; *Yu et al., 2012*; *Zhao et al., 2007*). Although initially identified as transcriptional coactivators, YAP/TAZ can also function as corepressors to inhibit target gene transcription by recruiting the nucleosome remodeling and histone deacetylase (NuRD) complex (*Kim et al., 2015b*). The evidence of Hippo pathway dysregulation in a variety of cancers and the list of YAP/TAZ target genes continue to increase (*Calses et al., 2019*; *Kulkarni et al., 2020*;

*Nguyen and Yi, 2019*; *Wang et al., 2018*; *Zanconato et al., 2016*). Dysregulation of YAP/TAZ-TEAD transcriptional output endows tumor cells with every hallmark of cancer, including sustained proliferation, resistance to apoptosis, tumor-promoting inflammation, tumor immune escape, dysregulated tumor metabolism, etc. (*Calses et al., 2019*; *Hanahan, 2022*; *Kulkarni et al., 2020*; *Nguyen and Yi, 2019*; *Zanconato et al., 2016*).

At the cellular organization level, the YAP/TAZ-TEAD transcription complex modulates mitochondrial fusion; cytoskeleton, primary cilium, and focal adhesion assembly; and caveolae formation (*Kim et al., 2015a*; *Mason et al., 2019*; *Nagaraj et al., 2012*; *Qiao et al., 2017*; *Rausch et al., 2019*). Processing bodies (P-bodies) are cytoplasmic membraneless organelles that consist of ribonucleoprotein complexes (RNPs) and are formed by phase separation (*Luo et al., 2018*; *Riggs et al., 2020*). Although initial studies hypothesized that mRNAs in P-bodies are targeted for decay and translational repression, it was subsequently suggested that P-bodies are not required for mRNA decay and that repressed mRNAs can be recycled from P-bodies to reenter translation; thus, the primary function of P-bodies is controlling the storage of untranslated mRNAs (*Decker and Parker, 2012*; *Hubstenberger et al., 2017*; *Luo et al., 2018*). The role of P-bodies in tumorigenesis and tumor progression is not well studied (*Anderson et al., 2015*; *Lavalée et al., 2021*; *Riggs et al., 2020*). The formation of P-bodies is correlated with epithelial–mesenchymal transition (EMT) in breast cancer (*Hardy et al., 2017*). In contrast, there is also evidence that an increase in P-bodies leads to attenuated growth, migration, and invasion of prostate cancer cells (*Bearss et al., 2021*). Recently, YAP was reported to be a negative regulator of P-bodies and to be involved in Kaposi sarcoma-associated herpesvirus (KHSV)-induced P-body disassembly in human umbilical vein endothelial cells (HUVECs) (*Castle et al., 2021*). However, this regulatory axis and the potential function of P-bodies in YAP-induced tumorigenesis remain unclear.

In this study, we discovered that YAP/TAZ are enhancers but not negative regulators of P-body formation in a series of cancer cell lines. YAP/TAZ modulates the transcription of multiple P-body-related genes, especially repressing the transcription of the tumor suppressor proline-rich nuclear receptor coactivator 1 (*PNRC1*) through cooperation with the NuRD complex. As a direct YAP/TAZ target gene, PNRC1 functions as a critical effector in YAP-induced biogenesis of P-bodies and tumorigenesis in colorectal cancer (CRC). Furthermore, disruption of P-bodies by knockdown of core component genes of P-bodies attenuated the protumorigenic effects of YAP in CRC. Thus, our study reveals a YAP–P-body positive regulatory axis in CRC, which exposes the vital role of YAP/TAZ in the biogenesis of P-bodies in tumors and implies that reexpression of PNRC1 or disruption of P-bodies is a potential therapeutic strategy for cancers with active YAP.

## Results
### YAP/TAZ regulates the transcription of P-body-related genes

Previously, to identify the new target genes and molecular signatures of YAP/TAZ in CRC, we performed RNA sequencing analysis of HCT116 CRC cells with simultaneous knockdown of YAP and TAZ (GSE176475) (*Guo et al., 2022*). Gene Ontology enrichment analysis of the 674 differentially expressed genes upon knockdown of YAP/TAZ (fold change [FC] > 2, p<0.05) revealed that the genes downregulated by YAP/TAZ knockdown were enriched in the term P-body in the cellular component category (*Figure 1A*). We further expanded our analysis to the moderately differentially expressed genes (FC < 0.66 or >1.5) that were annotated as related to P-bodies (*Figure 1—figure supplement 1A*, *Supplementary file 1*). Through integration with the public ChIP-seq data for TEAD4 in HCT116 cells from the ENCODE database, *AJUBA, WTIP, NOCT, SAMD4A,* and *PNRC1* were selected for in-depth investigation (*Figure 1B*, *Figure 1—figure supplement 1A*). Intriguingly, the public TEAD4 ChIP-seq datasets for the other three cancer cell lines (A549, MCF7, and MDA-MB-231), not just HCT116 cells, also showed strong TEAD4 binding peaks in the genomic loci of these five P-body-related genes (*Figure 1—figure supplement 1B*; *Mei et al., 2017*). HCT116, A549, MCF7, and MDA-MB-231 are well-established cell models for exploring YAP/TAZ function and the cell proliferation of these four cell lines is dependent on YAP/TAZ activity (*Rosenbluh et al., 2012*; *Shreberk-Shaked et al., 2020*; *Zanconato et al., 2015*; *Zhu et al., 2019*). It is worth noting that cell contact inhibition was observed in HCT116 and MDA-MB-231 and YAP remains in the nucleus regardless of cell–cell contact in A549 and MCF7 cells (*Fan et al., 2013*; *Kim et al., 2011*; *Lee et al., 2018*; *Wu*

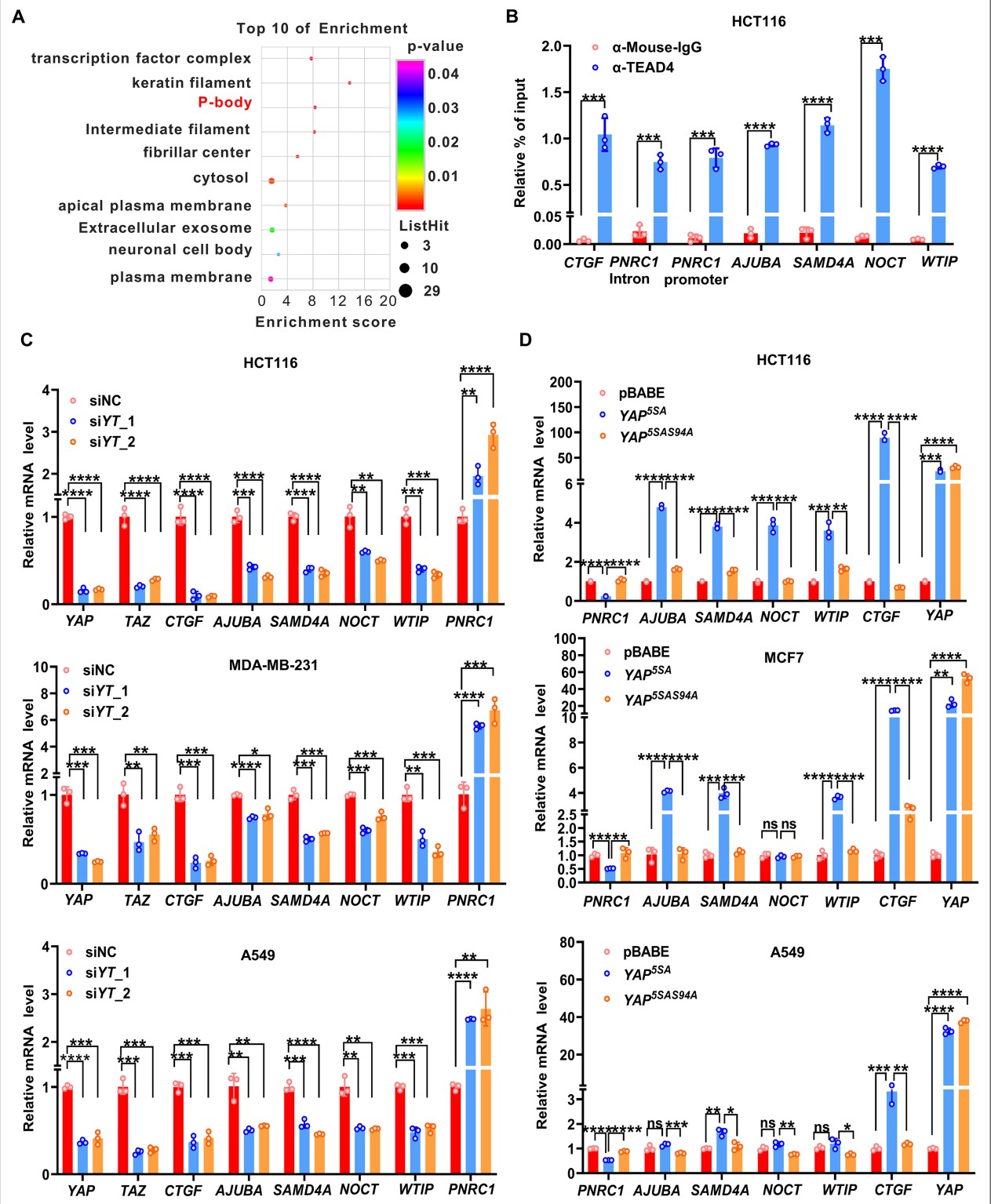

**Figure 1.** YAP/TAZ transcriptionally regulates genes related to P-bodies. (**A**) Gene Ontology (GO) analysis of the downregulated genes upon knockdown of *YAP/TAZ* in HCT116 cells. The graph shows enrichment in the cellular component category. (**B**) ChIP–qPCR analysis of endogenous TEAD4 binding to the genomic locus of the indicated P-body-related genes in HCT116 cells. The *CTGF* promoter was included as the positive control. (**C**) qPCR analysis of the mRNA levels of the indicated P-body-related genes in *YAP/TAZ* knockdown HCT116, MDA-MB-231, and A549 cells. Cells were transfected with *YAP/TAZ* siRNA for 3 d before qPCR analysis. (**D**) qPCR analysis of the mRNA levels of the indicated P-body-related genes in HCT116, MCF7, and A549 cells stably expressing YAP[5SA] and YAP[5SA-S94A]. Cells were infected with *YAP[5SA]*- and *YAP[5SA-S94A]*-containing retroviruses and selected with

*Figure 1 continued on next page*

*Figure 1 continued*

puromycin for 1 wk before qPCR analysis. n = 3 biologically independent samples per group. Two-tailed Student's *t*-test (**B**) and one-way ANOVA (**C, D**) were performed to assess statistical significance in this figure. These data (**B–D**) are representative of three independent experiments.

The online version of this article includes the following source data and figure supplement(s) for figure 1:

**Source data 1.** Original data for the statistical analysis in *Figure 1B–D*.

**Figure supplement 1.** RNA-seq and ChIP-seq analysis of YAP/TEAD's target genes related to P-bodies.

**Figure supplement 1—source data 1.** Original data for the heat map in *Figure 1—figure supplement 1A*.

**Figure supplement 1—source data 2.** Original file for the Western blot analysis in *Figure 1—figure supplement 1C and D*.

*et al., 2019*). The ChIP–qPCR results in HCT116 cells further confirmed that TEAD4 bound to the promoter regions of *AJUBA, WTIP, NOCT, SAMD4A,* and *PNRC1* and to the intronic region of *PNRC1* (*Figure 1B*). Next, we confirmed the significantly downregulated mRNA expression of *AJUBA, WTIP, SAMD4A,* and *NOCT* and moderately increased expression of *PNRC1* in YAP/TAZ knockdown HCT116 cells by qPCR analysis; this pattern was also observed in A549 lung cancer cells and MDA-MB-231 breast cancer cells (*Figure 1C*). Consistent with these findings, overexpression of the constitutively active YAP$^{5SA}$ mutant but not the TEAD binding-defective YAP$^{5SA-S94A}$ mutant significantly decreased the mRNA level of *PNRC1* and increased the mRNA level of *SAMD4A* in HCT116, MCF7 and A549 cells (*Figure 1D*). Enhanced expression of *AJUBA* and *WTIP* was observed in HCT116 and MCF7 cells but not in A549 cells (*Figure 1D*). Since *NOCT* was not affected by overexpression of YAP$^{5SA}$ in either MCF7 or A549 cells, we did not investigate *NOCT* in subsequent functional experiments (*Figure 1D*). Finally, we confirmed that the protein level of PNRC1 was increased by knockdown of YAP/TAZ in HCT116 cells (*Figure 1—figure supplement 1C*). Additionally, overexpression of YAP$^{5SA}$ but not YAP$^{5SA-S94A}$ decreased the protein level of PNRC1 in HCT116, A549, and MDA-MB-231 cells (*Figure 1—figure supplement 1D*). Overall, these data demonstrate that YAP/TAZ modulates the transcription of P-body-related genes through the TEAD transcription factors.

## YAP/TAZ enhances P-body formation

In contrast to stress granule (SG) formation, P-body formation is constitutive and independent of the activation of the integrated stress response (ISR) (*Luo et al., 2018*; *Riggs et al., 2020*). DEAD-box ATP-dependent RNA helicase 6 (DDX6) and mRNA-decapping enzyme 1A (DCP1A) are the essential components of P-bodies and are normally used as the biomarkers for P-bodies (*James et al., 2010*; *Lavalée et al., 2021*; *Luo et al., 2018*). To explore whether YAP/TAZ regulates P-body formation, we performed immunofluorescence analysis of DDX6 and DCP1A in YAP/TAZ knockdown cells plated at a low density. We found that knockdown of YAP/TAZ significantly decreased but overexpression of YAP$^{5SA}$ increased the number of DDX6/DCP1A-positive foci in HCT116 cells (*Figure 2A and B*). HCT116 cells expressing YAP$^{5SA-S94A}$ and control HCT116 cells showed similar numbers of P-bodies, which indicated that the TEAD transcription factors mediate the enhanced effects of YAP/TAZ on P-body formation (*Figure 2B*). Similar results were observed in YAP/TAZ knockdown MDA-MB-231 cells and YAP$^{5SA}$/YAP$^{5SA-S94A}$-expressing A549 cells (*Figure 2—figure supplement 1A and B*).

YAP/TAZ are well known to be activated by serum stimulation and suppressed by high cell densities (*Yu et al., 2012*; *Zhao et al., 2007*). Of note, cytoplasmic translocation of YAP at high cell density was first observed in the untransformed NIH3T3 cells (*Zhao et al., 2007*). Thus, in addition to a series of cancer cell lines, NIH3T3 cells were further included in this study. Consistently, overexpression of YAP$^{5SA}$ but not the YAP$^{5SA-S94A}$ increased the number of DDX6/LSM14A-positive foci in NIH3T3 cells (*Figure 2—figure supplement 2A*). Upregulation of *Ajuba, Samd4* (mouse ortholog of human *SAMD4A*) and *Noct* and downregulation of *Pnrc1* was also observed in NIH3T3 cells overexpressed with YAP$^{5SA}$ but not cells with YAP$^{5SA-S94A}$ overexpression (*Figure 2—figure supplement 2E*). More-over, despite their constitutive formation in cells, the size and number of P-bodies are altered in response to stress (*Luo et al., 2018*; *Riggs et al., 2020*). Next, we evaluated P-bodies under exposure to different stimuli. We observed that serum stimulation led to rapid induction of P-body formation in HCT116 and NIH3T3 cells (*Figure 2C*, *Figure 2—figure supplement 2B*). Knockdown of YAP/TAZ attenuated the enhancement of P-body formation induced by serum stimulation (*Figure 2—figure supplement 1C*). Conversely, at a high cell density, the number of P-bodies was significantly decreased in HCT116 and NIH3T3 cells (*Figure 2D*, *Figure 2—figure supplement 2C*). Consistent

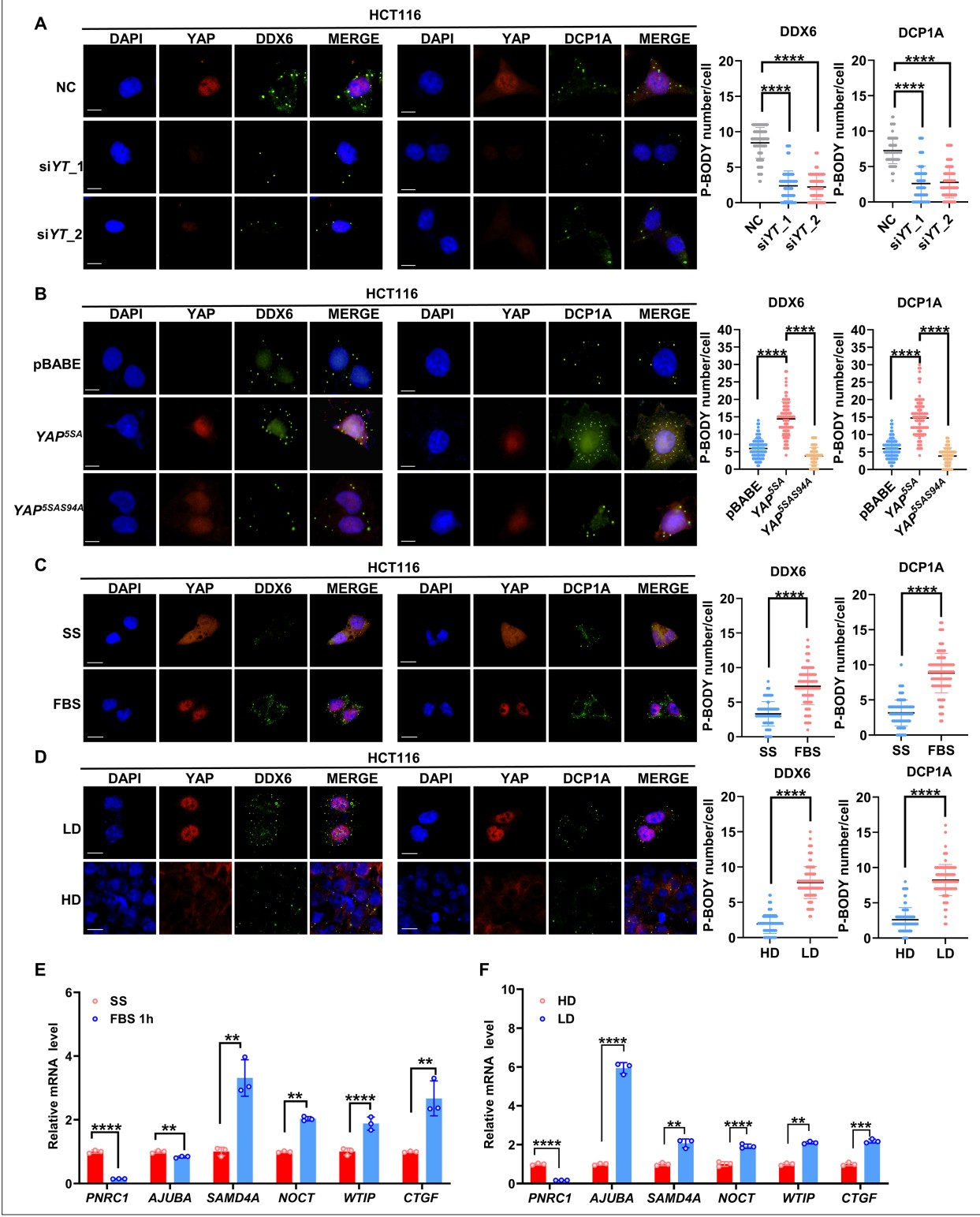

**Figure 2.** YAP/TAZ promotes P-body formation in colorectal cancer (CRC) cells. (**A**) Immunofluorescence analysis of the P-body markers DDX6 and DCP1A in *YAP/TAZ* knockdown HCT116 cells. Cells were transfected with *YAP/TAZ* siRNA for 3 d before processing for immunofluorescence staining using anti-DDX6 and anti-DCP1A antibodies. Foci were counted in 100 cells per group. (**B**) Immunofluorescence analysis of DDX6 and DCP1A in HCT116 cells expressing YAP[5SA] and YAP[5SA-S94A]. (**C**) Immunofluorescence analysis of DDX6 and DCP1A in HCT116 cells. Cells were treated with 10% fetal bovine serum (FBS) for 1 hr after overnight serum starvation (SS). (**D**) Immunofluorescence analysis of DDX6 and DCP1A in HCT116 cells in sparse or confluent culture. (**E, F**) qPCR analysis of the indicated genes in HCT116 cells. HCT116 cells were treated with 10% FBS for 1 hr after overnight SD (**E**) or culture

*Figure 2 continued on next page*

*Figure 2 continued*

under sparse or confluent conditions in standard culture medium (**F**). Kruskal–Wallis test (**A, B**), Mann–Whitney *U* test (**C, D**), and two-tailed Student's *t*-test (**E, F**) were performed to assess statistical significance. These data (**A–F**) are representative of three independent experiments.

The online version of this article includes the following source data and figure supplement(s) for figure 2:

**Source data 1.** Original data for the statistical analysis in *Figure 2A–F*.

**Figure supplement 1.** YAP/TAZ modulates P-body formation in breast, lung and colorectal cancer cells.

**Figure supplement 1—source data 1.** Original data for the statistical analysis in *Figure 2—figure supplement 1A–C*.

**Figure supplement 2.** YAP/TAZ modulates P-body formation in untransformed NIH3T3 cells.

**Figure supplement 2—source data 1.** Original data for the statistical analysis in *Figure 2—figure supplement 2A–H*.

with this finding, the expression of *SAMD4A*, *NOCT*, and *WTIP* in HCT116 cells was induced by serum stimulation and suppressed by culture at a high cell density (*Figure 2E and F*). Similar results were also observed in NIH3T3 cells (*Figure 2—figure supplement 2F and G*). Intriguingly, both serum starvation and culture at a high cell density dramatically increased the expression of *PNRC1*, consistent with the tumor suppressor function of PNRC1 (*Figure 2E and F*, *Figure 2—figure supplement 2F and G*). Recent studies have revealed that mechanical cues as an important signal modulating YAP/TAZ activity (*Aragona et al., 2013*; *Dupont et al., 2011*). Diverse mechanical forces, such as increased extracellular matrix (ECM) rigidity, cell stretching, shear stress, or the increased area of cell adhesion, can all activate YAP, which is dominant over Hippo signaling (*Dasgupta and McCollum, 2019*; *Piccolo et al., 2014*). Next, we examined whether ECM stiffness affected P-body formation. When NIH3T3 cells were shifted from soft (1 kPa) to stiff (40 kPa) matrices, YAP was translocated into nucleus and activated (*Figure 2—figure supplement 2D*). Furthermore, the P-body formation was enhanced, which was associated with decreased mRNA level of *Pnrc1* and increased mRNA levels of *Ajuba, Samd4,* and *Noct* (*Figure 2—figure supplement 2D and H*). Collectively, our data indicate that YAP/TAZ could be positive regulators of P-body formation in response to various stimuli, probably by modulating the expression of P-body-related genes.

## SAMD4A, AJUBA, and PNRC1 mediate the functions of YAP/TAZ in regulating P-body formation

Next, we investigated whether the P-body-related genes transcriptionally regulated by YAP/TAZ mediate the biological functions of YAP/TAZ in regulating P-body formation. The LIM-domain proteins AJUBA, WTIP, and LIMD1 are known as negative regulators of LATS1 (*Das Thakur et al., 2010*). They are also components of P-bodies and are required for miRNA-mediated silencing (*James et al., 2010*). SAMD4A is the mammalian homolog of *Drosophila* Smaug, which is involved in translational repression and localized in P-bodies (*Baez and Boccaccio, 2005*). First, we knocked down *AJUBA* and *SAMD4A* in HCT116 cells overexpressing YAP⁵ˢᴬ (*Figure 3—figure supplement 1A and B*). As expected, knockdown of both *AJUBA* and *SAMD4A* significantly diminished the promoting effect of YAP⁵ˢᴬ overexpression on P-body formation in HCT116 cells (*Figure 3A*). Unlike AJUBA and SAMD4A, PNRC1 is a tumor suppressor that inhibits P-body formation by recruiting cytoplasmic DCP1A/DCP2 into the nucleolus, thus loss of cytoplasmic DCP1A/DCP2 results in disruption of P-body (*Gaviraghi et al., 2018*). Overexpression of YAP⁵ˢᴬ suppressed PNRC1 expression; thus, WT PNRC1 and PNRC1 with the W300A mutation, which disrupts the interaction between PNRC1 and DCP1A/DCP2, were overexpressed in YAP⁵ˢᴬ-expressing HCT116 cells (*Figure 3—figure supplement 1C and D*; *Gaviraghi et al., 2018*). We observed that overexpression of WT PNRC1 but not the W300A mutant dramatically decreased the number of P-bodies in YAP⁵ˢᴬ-expressing HCT116 cells (*Figure 3B*). We also examined whether the attenuation of P-body formation by knockdown of YAP/TAZ can be restored by knockdown of *PNRC1* (*Figure 3—figure supplement 1E*). Consistent with the above findings, the reduction in the P-body number was reversed by knockdown of *PNRC1* in YAP/TAZ knockdown HCT116 cells (*Figure 3—figure supplement 1F*). Collectively, these findings indicate that YAP/TAZ enhances P-body formation through modulation of a series of P-body-related genes. Both activation of SAMD4A and AJUBA expression and downregulation of PNRC1 are involved in YAP/TAZ-induced P-body formation.

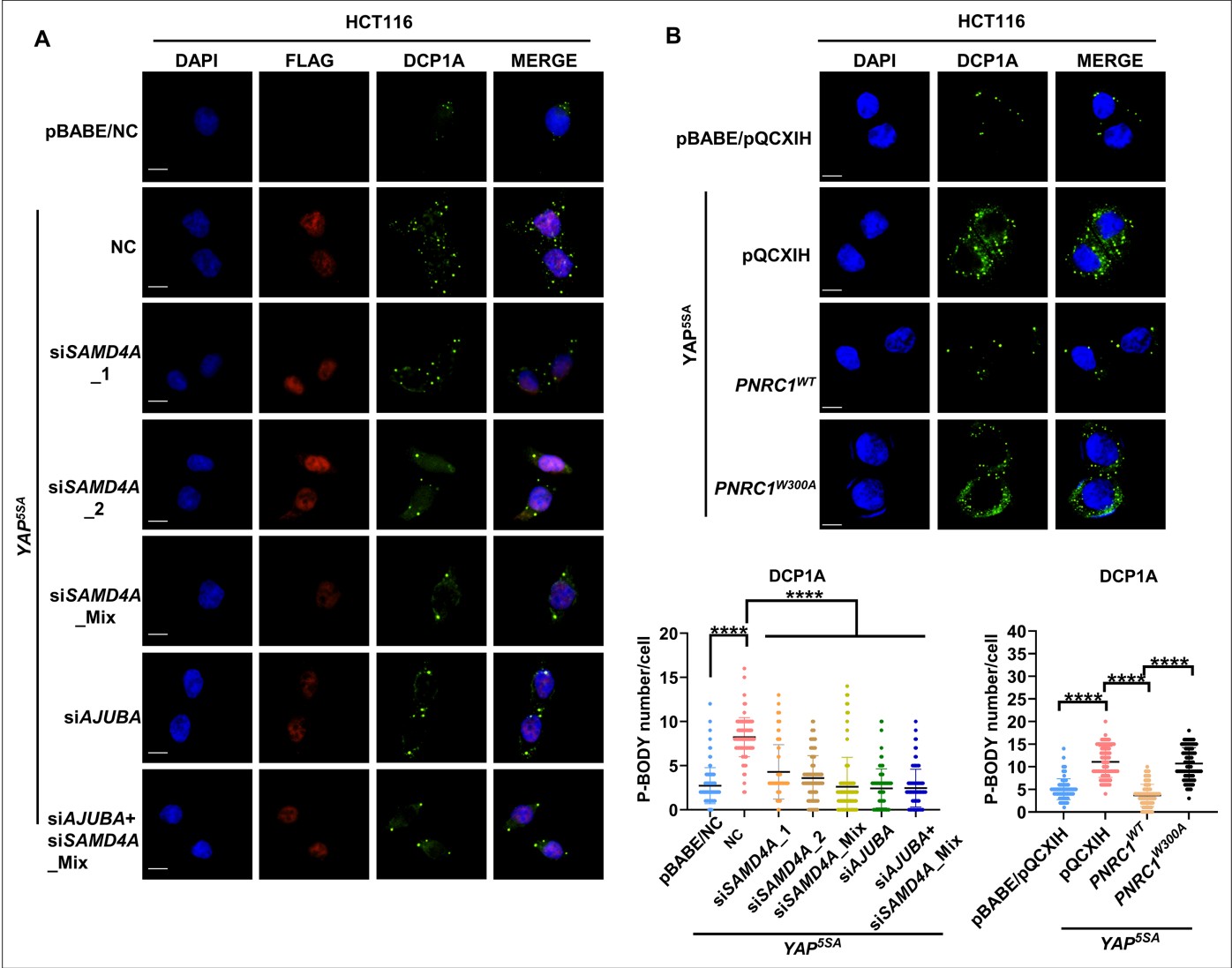

**Figure 3.** SAMD4A, AJUBA, and PNRC1 mediate the regulatory functions of YAP/TAZ in P-body formation. (**A**) Immunofluorescence analysis of DDX6 and DCP1A in HCT116 cells stably expressing YAP[5SA] and YAP[5SA]-expressing cells transiently transfected with *SMAD4A* and *AJUBA* siRNA. Foci were counted in 100 cells per group. (**B**) Immunofluorescence analysis of DDX6 and DCP1A in HCT116 cells expressing YAP[5SA] alone or in combination with PNRC1[WT] or PNRC1[W300A]. Kruskal–Wallis test was performed to assess statistical significance. These data (**A–B**) are representative of three independent experiments.

The online version of this article includes the following source data and figure supplement(s) for figure 3:

**Source data 1.** Original data for the statistical analysis in *Figure 3A and B*.

**Figure supplement 1.** Knockdown of *PNRC1* reverses the attenuated P-body formation induced by *YAP/TAZ* knockdown.

**Figure supplement 1—source data 1.** Original data for the statistical analysis in *Figure 3—figure supplement 1A–F*.

**Figure supplement 1—source data 2.** Original file for the western blot analysis in *Figure 3—figure supplement 1D*.

## YAP/TAZ inhibit PNRC1 gene transcription by recruiting the NuRD complex

PNRC1 is a newly identified tumor suppressor gene whose expression is frequently downregulated in cancer (*Gaviraghi et al., 2018*). Thus, we further explored the molecular mechanism of YAP/TAZ in inhibiting the *PNRC1* gene transcription. The ChIP-seq data for TEAD4 at the *PNRC1* gene locus in multiple cancer cells implicated PNRC1 as a potential direct target gene of YAP/TAZ-TEAD transcription complexes (*Figure 1—figure supplement 1B*). As ChIP–qPCR analysis of TEAD4 in HCT116 cells revealed one TEAD4 binding site at the *PNRC1* promoter and another in the *PNRC1* intron, we

constructed *PNRC1* promoter and *PNRC1* intron luciferase reporter plasmids. We observed that over-expression of YAP[5SA] significantly decreased the luciferase activity of both the *PNRC1* promoter and intron reporters (*Figure 4A*). Compared with the 5SA mutation in YAP, the S94A mutation resulted in a decreased suppressive effect on *PNRC1* promoter and intron luciferase reporter activity (*Figure 4A*). In contrast, the luciferase activity of both the *PNRC1* promoter and intron reporters was significantly enhanced in *YAP/TAZ* knockdown HCT116 cells (*Figure 4B*, *Figure 4—figure supplement 1A*). Bioin-formatic analysis of TEAD4 ChIP peaks in the *PNRC1* promoter and intronic regions with JASPAR revealed the existence of one TEAD binding motif in each peak region; thus, we further constructed *PNRC1* luciferase reporter plasmids with mutated TEAD binding sites. Consistent with the above results, mutation of the TEAD binding sites abolished the inhibitory effect of YAP[5SA] on the *PNRC1* promoter and intron luciferase reporters (*Figure 4C*). Similarly, mutation of the TEAD binding sites escaped the derepression of *PNRC1* promoter and intron luciferase reporters by *YAP/TAZ* knockdown (*Figure 4D*). Furthermore, the ChIP–qPCR results confirmed that YAP bound to the promoter and intronic regions of *PNRC1*, which required its interaction with TEADs (*Figure 4—figure supplement 1B and C*).

In addition to functioning as transcriptional coactivators, YAP/TAZ can also act as transcriptional corepressors by recruiting the NuRD complex (*Kim et al., 2015b*). The ChIP–qPCR results showed that the NuRD complex component CHD4 was recruited to the promoter and intronic regions of the *PNRC1* gene by overexpressed YAP[5SA] but not by the TEAD binding-defective YAP[5SA-S94A] mutant (*Figure 4E*). Compared to the genomic locus of *PNRC1*, the binding enrichment of CHD4 at the YAP target genes activated by YAP/TAZ was relatively lower and not affected by overexpression of YAP (*Figure 4—figure supplement 1D*). Moreover, knockdown of the NuRD complex components CHD4 and RBBP4 significantly upregulated the mRNA expression of *PNRC1* in HCT116 cells (*Figure 4F*). Consistently, knockdown of *CHD4* significantly decreased the number of DDX6/DCP1A-positive foci in HCT116 cells (*Figure 4—figure supplement 1E*). Taken together, these data demonstrate that YAP/TAZ inhibits *PNRC1* gene transcription through direct binding of TEADs to the PNRC1 gene locus and that the NuRD complex is required for the transcriptional repression of *PNRC1* by YAP/TAZ.

## PNRC1 suppresses the oncogenic function of YAP in CRC

Analysis of colorectal (COAD) and rectal (READ) TCGA datasets revealed that the mRNA level of PNRC1 was significantly decreased in CRC (*Figure 5—figure supplement 1A*). We further confirmed the decreased mRNA level of *PNRC1* in CRC by qPCR analysis of 16 CRC tissues with paired normal mucosal tissues; this finding implies that PNRC1 is a potential tumor suppressor also in CRC (*Figure 5—figure supplement 1B*). Thus, we sought to explore whether downregulation of PNRC1 mediates the oncogenic function of YAP in CRC. To this end, we examined whether the YAP overexpression-induced oncogenic phenotype can be attenuated by coexpression of YAP[5SA] with WT PNRC1 or the W300A mutant in HCT116 cells. We observed that reexpression of WT PNRC1 almost completely abolished the increases in cell proliferation and colony formation induced by YAP[5SA] overexpression in HCT116 cells (*Figure 5A and B*). Re-expression of the PNRC1 W300A mutant did not affect the proliferation and colony formation of YAP[5SA]-expressing HCT116 cells, which implied that the suppressive effect of PNRC1 on YAP relies on the recruitment of cytoplasmic DCP1A/DCP2 into the nucleolus by PNRC1 (*Figure 5A and B*). Similarly, overexpression of PNRC1 WT but not PNRC1 W300A suppressed the increase in migration induced by YAP[5SA] in HCT116 cells (*Figure 5C*). To verify the tumor-suppressive effect of PNRC1 on YAP in CRC in vivo, we performed a xenograft assay by subcutaneously injecting HCT116 cells into nude mice. Consistent with the above findings, reexpression of WT PNRC1 but not the W300A mutant dramatically inhibited the growth of YAP[5SA]-expressing HCT116 xenografts, and xenograft tumors formed from HCT116 cells coexpressing YAP[5SA] and PNRC1 were significantly smaller than the tumors formed from HCT116 cells expressing YAP[5SA] alone or in combination with the PNRC1 W300A mutant (*Figure 5D*). Ki67 staining of xenograft tumors further showed fewer Ki67-positive cells in xenograft tumors formed from HCT116 cells coexpressing YAP[5SA] and PNRC1 (*Figure 5E*, *Figure 5—figure supplement 1C*). Next, we examined whether the *YAP/TAZ* knockdown-induced attenuation of the oncogenic pheno-type can be restored by knockdown of *PNRC1* in HCT116 cells. Intriguingly, the decrease in prolifer-ation and attenuation of migration induced by *YAP/TAZ* knockdown were reversed by knockdown of *PNRC1* in HCT116 cells (*Figure 5—figure supplement 1D and E*). Overall, these results indicate that

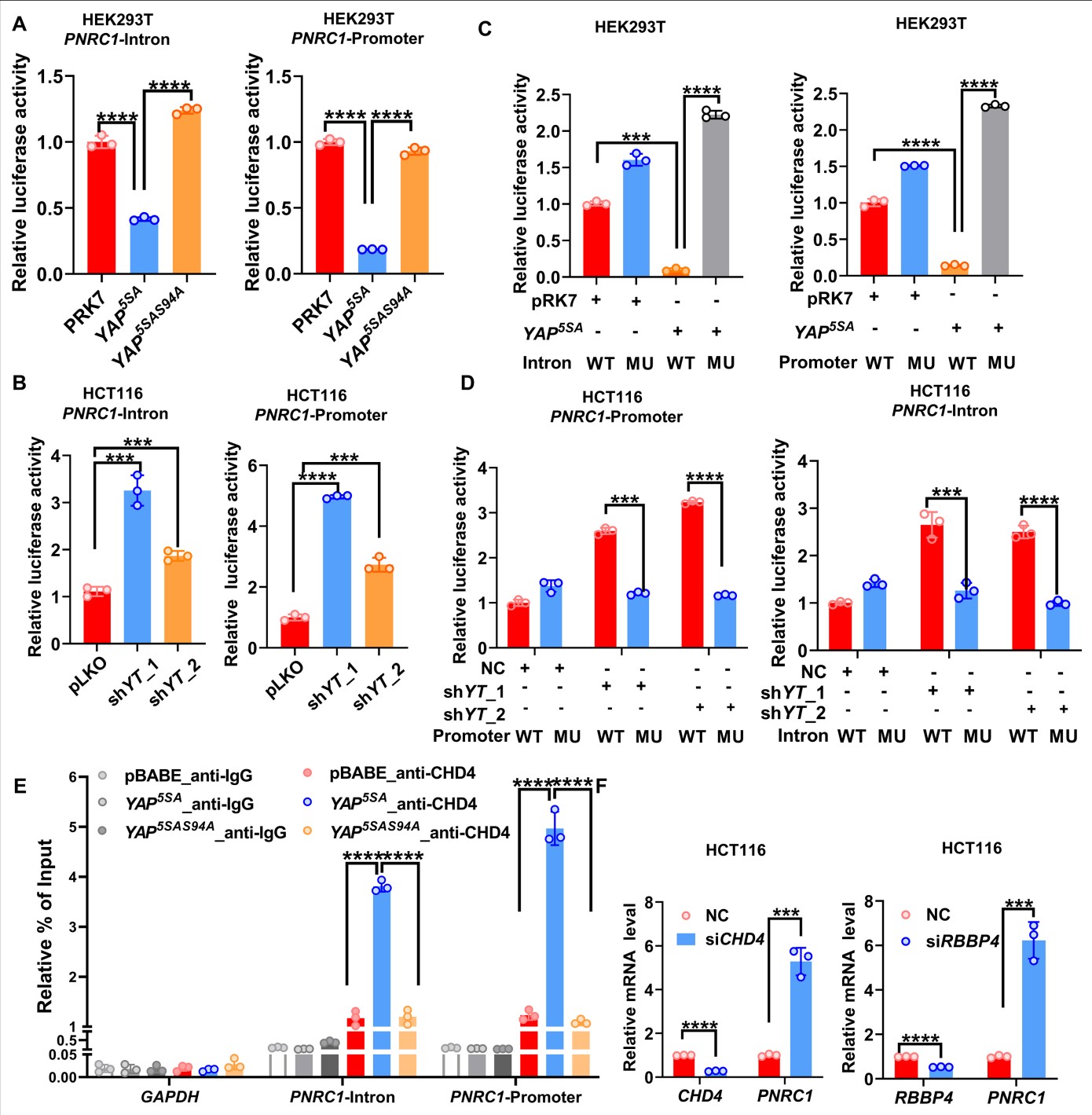

**Figure 4.** YAP suppresses PNRC1 gene transcription by recruiting the NuRD complex. (**A**) Overexpression of YAP[5SA] but not YAP[5SA-S94A] decreased the luciferase activity of the *PNRC1* promoter and intron reporters. HEK-293T cells were transfected with the indicated FLAG-*YAP[5SA]* and *YAP[5SA-S94A]* expression plasmids and the *PNRC1* promoter or intron luciferase reporter. (**B**) Knockdown of *YAP/TAZ* stimulated the luciferase activity of the *PNRC1* promoter and intron reporters. The *PNRC1* promoter or intron luciferase reporter plasmid and the Renilla luciferase reporter plasmid were co-transfected into HCT116 cells stably expressing pLKO-vec, sh*YAP/TAZ*-1, or sh*YAP/TAZ*-2. (**C, D**) Luciferase assay of the *PNRC1* promoter/intron WT reporters and mutant reporters with TEAD binding motif mutations in HEK-293T cells (**C**) and HCT116 cells (**D**). (**E**) ChIP–qPCR analysis of CHD4 binding to the *PNRC1* promoter and intronic regions in control and HCT116 cells stably expressing FLAG-YAP[5SA] or YAP[5SA-S94A]. (**F**) qPCR analysis of *PNRC1, CHD4,* and *RBBP4* in HCT116 cells transfected with the indicated siRNAs. n = 3 biologically independent samples per group. One-way ANOVA (**A–E**) and two-tailed Student's *t*-test (**F**) were performed to assess statistical significance in this figure. These data (**A–F**) are representative of two independent experiments.

*Figure 4 continued on next page*

*Figure 4 continued*

The online version of this article includes the following source data and figure supplement(s) for figure 4:

**Source data 1.** Original data for the statistical analysis in *Figure 4A–F*.

**Figure supplement 1.** TEADs and CHD4 mediates YAP-dependent inhibition on *PNRC1* gene transcription.

**Figure supplement 1—source data 1.** Original data for the statistical analysis in *Figure 4—figure supplement 1A–E*.

**Figure supplement 1—source data 2.** Original file for the western blot analysis in *Figure 4—figure supplement 1A*.

YAP promotes tumorigenesis by downregulating PNRC1 expression and that reexpression of PNRC1 suppresses YAP-driven tumor growth.

## P-body disassembly attenuates YAP-driven cell proliferation and migration in CRC

Due to the inhibitory effects of PNRC1 on P-body formation and the YAP-induced oncogenic phenotype, we evaluated whether enhanced P-body formation plays a vital role in YAP-driven cancer cell proliferation and migration. The proteins LSM14 homolog A (LSM14A) and DDX6 are essential nucleating proteins for P-body assembly, and DCP1A plays a vital role in further RNP aggregation, which is required for stress-dependent P-body aggregation (*Lavalée et al., 2021*; *Luo et al., 2018*; *Riggs et al., 2020*). To explore the requirement of P-body formation for the YAP-induced oncogenic phenotype, we generated YAP$^{5SA}$-expressing HCT116 cell lines with stable knockdown of *DCP1A*, *LSM14A*, and *DDX6* (*Figure 6—figure supplement 1A*, *Figure 6—figure supplement 2A–C*). By immunofluorescence analysis of DDX6 and DCP1A, we further confirmed the knockdown of DCP1A and DDX6 and observed a reduced number of P-bodies upon knockdown of DCP1A, LSM14A or DDX6 in YAP$^{5SA}$-expressing HCT116 cells (*Figure 6—figure supplement 1B*). Next, by using a CCK8 assay, we found that knockdown of *DCP1A*, *LSM14A*, and *DDX6* suppressed the proliferation of YAP$^{5SA}$-expressing and control 'parental' HCT116 cells, consistent with the results of the colony formation assay (*Figure 6A and B*). As an oncogene, YAP is known to promote cell division and inhibit cell apoptosis of cancer cells (*He et al., 2020*; *Jang et al., 2017*). By analyzing the cell cycle and cell apoptosis, we further found that knockdown of *DCP1A, LSM14A,* and *DDX6* all led to downregulation of cell mitosis and increased cell apoptosis, which was opposite to the effect of YAP$^{5SA}$ overexpression in HCT116 cells (*Figure 6—figure supplement 3A and B*). Furthermore, knockdown of either *DCP1A* or *LSM14A* significantly attenuated the enhancement of cell migration induced by overexpression of YAP$^{5SA}$ in HCT116 cells (*Figure 6C*). In contrast, knockdown of *DDX6* showed stimulative effect on the migration of both control and YAP$^{5SA}$-expressing HCT116 cells, possibly due to the diverse functions of DDX6 (*Figure 6—figure supplement 4A*; *Di Stefano et al., 2019*). To further demonstrate the potential role of P-body mediating the function of YAP/TAZ in CRC, we established YAP$^{5SA}$-expressing HCT116 cell lines with stable knockdown of *AJUBA* and *SAMD4A* (*Figure 6—figure supplement 4B and C*). Indeed, both knockdown of *AJUBA* and *SAMD4A* suppressed the proliferation and cell migration of YAP$^{5SA}$-expressing and control 'parental' HCT116 cells (*Figure 6—figure supplement 4D and E*). Collectively, our data demonstrate that P-body formation is required for the oncogenic function of YAP in CRC.

## Codependency of YAP/TEAD and essential P-body-related genes in pancancer CRISPR screens

Based on the observation that P-body disassembly attenuates YAP-driven cell proliferation in CRC cells, we speculated that cancer cells whose proliferation is dependent on YAP should also be vulnerable to knockout of essential P-body genes. To this end, we analyzed the Cancer Dependency Map (DepMap), which aims to systematically assess the effect of single-gene inactivation on cell proliferation by CRISPR and shRNA screens and define genetic dependencies in hundreds of cancer cell lines by integrating data pertaining to multiple molecular characteristics, such as Cancer Cell Line Encyclopedia (CCLE) data (*Dempster et al., 2021*; *Tsherniak et al., 2017*). As expected, by analyzing gene expression data from the CCLE, we observed a strong positive correlation between YAP-regulated P-body-related genes (*SAMD4A, AJUBA,* and *WTIP*) and canonical target genes of YAP (*CTGF, CYR61, AXL,* and *AMOTL2*) in cell lines across cancers or in cell lines of colorectal, breast and lung lineages (*Figure 7A*, *Figure 7—figure supplement 1A and B*). IHC analysis of 294 CRC tissues further showed

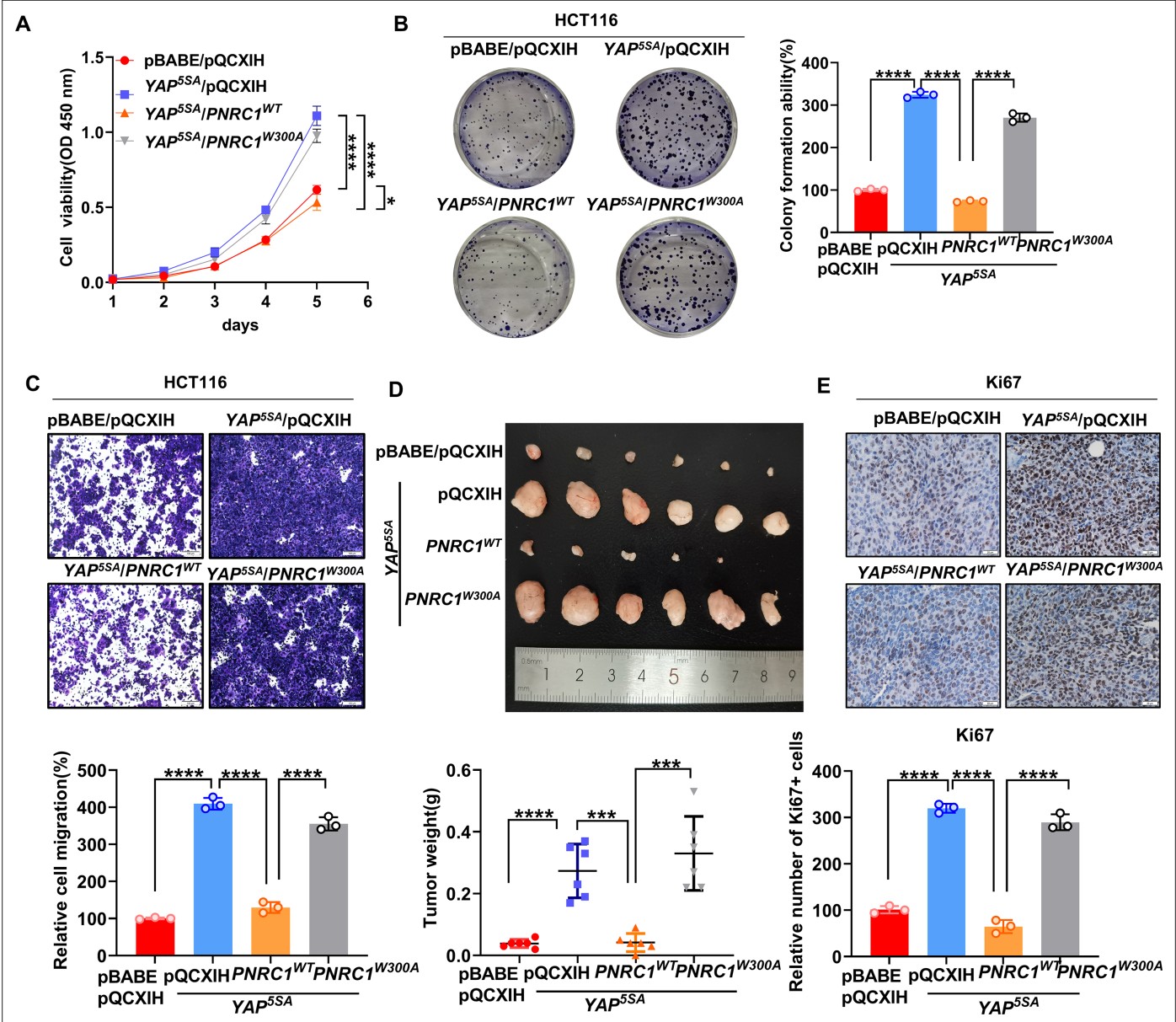

**Figure 5.** PNRC1 attenuates the oncogenic function of YAP in colorectal cancer (CRC). (**A**) CCK8 proliferation assays of HCT116 cells stably expressing YAP5SA alone or in combination with of PNRC1WT or PNRC1W300A. n = 4 biologically independent samples per group. (**B, C**) Colony formation assay (**B**) and Transwell assay (**C**) of HCT116 cells stably expressing YAP5SA alone or in combination with PNRC1WT or PNRC1W300A. n = 3 biologically independent samples per group. (**D**) Representative images of xenograft tumors formed from HCT116 cells stably expressing YAP5SA alone or in combination with of PNRC1WT or PNRC1W300A (n = 6). (**E**) Representative images of IHC staining of the proliferation marker Ki67 in xenograft tumors formed from HCT-116 cells stably expressing YAP5SA alone or in combination with PNRC1WT or PNRC1W300A (n = 3). Two-way ANOVA (**A**) and one-way ANOVA (**B–E**) were performed to assess statistical significance in this figure. These data (**A–C**) are representative of two independent experiments.

The online version of this article includes the following source data and figure supplement(s) for figure 5:

**Source data 1.** Original data for the statistical analysis in *Figure 5A–E*.

**Figure supplement 1.** *PNRC1* is a tumor suppressor gene in CRC.

**Figure supplement 1—source data 1.** Original data for the statistical analysis in *Figure 5—figure supplement 1B–E*.

the positive correlation between the expression of AJUBA/SAMD4A and YAP (*Figure 7—figure supplement 1D*). Although there were no correlations between PNRC1 and YAP target genes in cell lines across cancers, we found that the mRNA level of *PNRC1* was negatively correlated with that of YAP target genes in cancer cells of thyroid and central nervous system (CNS) lineages (*Figure 7B*,

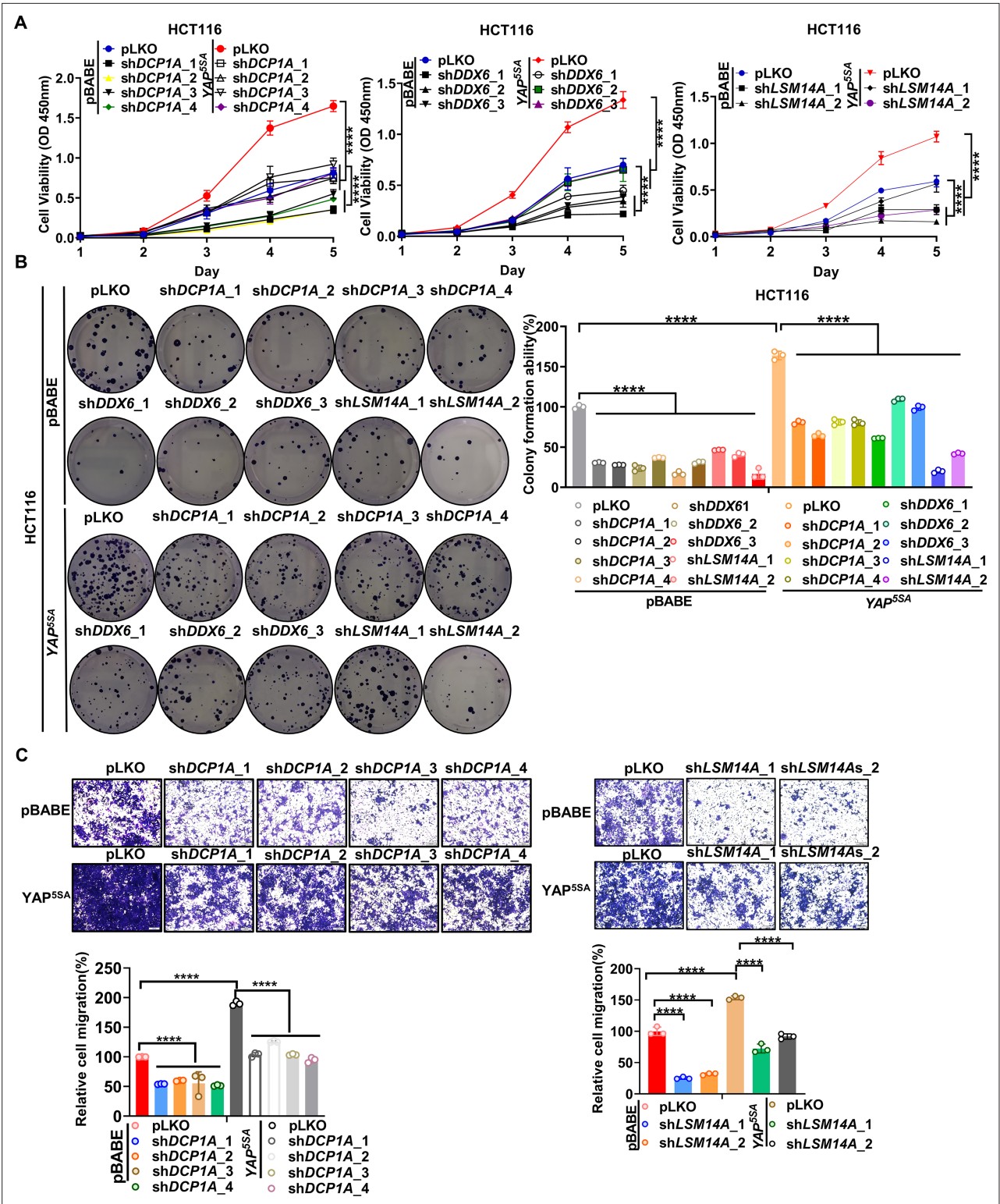

**Figure 6.** Knockdown of P-body-related core genes suppresses the oncogenic function of YAP in colorectal cancer (CRC). (**A**) CCK8 proliferation assays of control HCT116 cells with or without knockdown of *DCP1A, LSM14A* or *DDX6* and HCT116 cells stably expressing YAP[5SA] with or without knockdown of *DCP1A, LSM14A,* or *DDX6*. n = 5 biologically independent samples per group. (**B, C**) Colony formation assay (**B**) and Transwell assay (**C**) of control HCT116 cells with or without knockdown of *DCP1A, LSM14A,* or *DDX6* and HCT116 cells stably expressing YAP[5SA] with or without knockdown of *DCP1A, LSM14A,* or *DDX6*. n = 3 biologically independent samples per group. Two-way ANOVA (**A**) and one-way ANOVA (**B, C**) were performed to assess statistical significance in this figure. These data (**A–C**) are representative of three independent experiments.

*Figure 6 continued on next page*

*Figure 6 continued*

The online version of this article includes the following source data and figure supplement(s) for figure 6:

**Source data 1.** Original data for the statistical analysis in *Figure 6A–C*.

**Figure supplement 1.** Knockdown efficiency and P-body reduction in *DCP1A/LSM14A/DDX6* knockdown HCT116 cells.

**Figure supplement 1—source data 1.** Original data for the statistical analysis in *Figure 6—figure supplement 1A and B*.

**Figure supplement 1—source data 2.** Original file for the western blot analysis in *Figure 6—figure supplement 1A*.

**Figure supplement 2.** Confirmation of *DCP1A/ LSM14A/DDX6* knockdown in YAP⁵ˢᴬ-expressing HCT116 cells.

**Figure supplement 2—source data 1.** Original data for the statistical analysis in *Figure 6—figure supplement 2A–C*.

**Figure supplement 2—source data 2.** Original file for the western blot analysis in *Figure 6—figure supplement 2A–C*.

**Figure supplement 3.** Knockdown of *DCP1A, LSM14A* and *DDX6* downregulates cell mitosis and increases cell apoptosis in YAP⁵ˢᴬ-expressing HCT116 cells.

**Figure supplement 3—source data 1.** Original data for the bar plot in *Figure 6—figure supplement 3A and B*.

**Figure supplement 4.** Knockdown of *AJUBA* or *SAMD4A* attenuates the oncogenic function of YAP in HCT116 cells.

**Figure supplement 4—source data 1.** Original data for the statistical analysis in *Figure 6—figure supplement 4A–D*.

**Figure supplement 4—source data 2.** Original file for the western blot analysis in *Figure 6—figure supplement 4C*.

*Figure 7—figure supplement 1C*). Next, we analyzed the effect of P-body core gene knockout in 1070 cancer cell lines (DepMap 22Q1 Public+Score, Chronos). Strikingly, knockout of the P-body-nucleation-determining genes *DDX6* and *LSM14A* inhibited proliferation in multiple cancer cell lines (negative Chronos score) (*Figure 7C*). Similar results were observed for *EDC4*, which is required for P-body aggregation (*Figure 7C*). Logically, correlations between dependency profiles suggest functionality in the same pathway or regulatory axis; thus, *EDC4* is strongly associated with multiple known P-body genes (*DDX6, DCP2, EIF4ENIF1*, etc.) (*Supplementary file 2*). Furthermore, we found that *YAP* ranked 14th among genes correlating with *EDC4* (*Figure 7C*, *Supplementary file 2*). In addition, the YAP dependency score was positively correlated with the DDX6 and LSM14A scores (*Figure 7C*). Similar relationships were observed between EDC4/DDX6/LSM14A and TEAD1/3 (*Figure 7—figure supplement 2*). Last, we examined whether cell proliferation and cell migration are affected by the knockdown of *DCP1A* or *LSM14A* and overexpression of PNRC1 in MCF7, MDA-MB-231, and A549 cell lines, whose proliferation is dependent on YAP/TAZ activity. Consistent with the observation in HCT116 cells, the knockdown of *DCP1A/LSM14A* and overexpression of PNRC1 attenuated both cell proliferation and cell migration in these three YAP-dependent cancer cells (*Figure 7—figure supplements 3–5*). Collectively, the co-dependencies of YAP/TEAD and essential P-body genes further suggest that enhanced P-body formation plays a vital role in YAP-induced tumorigenesis.

## Discussion

Dysregulation of the Hippo pathway occurs in a variety of cancers, leading to cell transformation and diverse changes in tumor cells through activation of the YAP/TEAD transcriptional program (*Calses et al., 2019*; *Kulkarni et al., 2020*; *Nguyen and Yi, 2019*; *Wang et al., 2018*; *Zanconato et al., 2016*). Here, we demonstrated the crucial role of YAP/TEAD in regulating P-body formation in multiple cancer cell lines. Through transcriptional stimulation of positive regulators of P-body formation (AJUBA, WTIP, and SAMD4A) and suppression of negative regulators of P-body formation (PNRC1), YAP enhances P-body formation and increases the number of P-bodies in cancer cells, which suggests that YAP is a positive regulator of P-body formation (*Figure 7D*). Studies of P-bodies in yeast have shown that the size and number of P-bodies increase upon exogenous and endogenous stress (*Luo et al., 2018*). In addition, our study revealed that the number of P-bodies decreases under serum starvation, contact inhibition, and decreased ECM rigidity, possibly due to inactivation of YAP/TEAD. In contrast, a recent study, which provided the first link between YAP and P-bodies, implicated YAP as a negative regulator of P-bodies in KHSV-infected HUVECs (*Castle et al., 2021*). Elizabeth L. Castle et al. reported that virus-encoded Kaposin B (KapB) induces actin stress fiber formation and disassembly of P-bodies, which requires RhoA activity and the YAP transcriptional program (*Castle et al., 2021*). YAP-enhanced autophagic flux was proposed to participate in KapB-induced P-body disassembly, consistent with the concept that SGs and P-bodies are cleared by autophagy (*Buchan et al., 2013*;

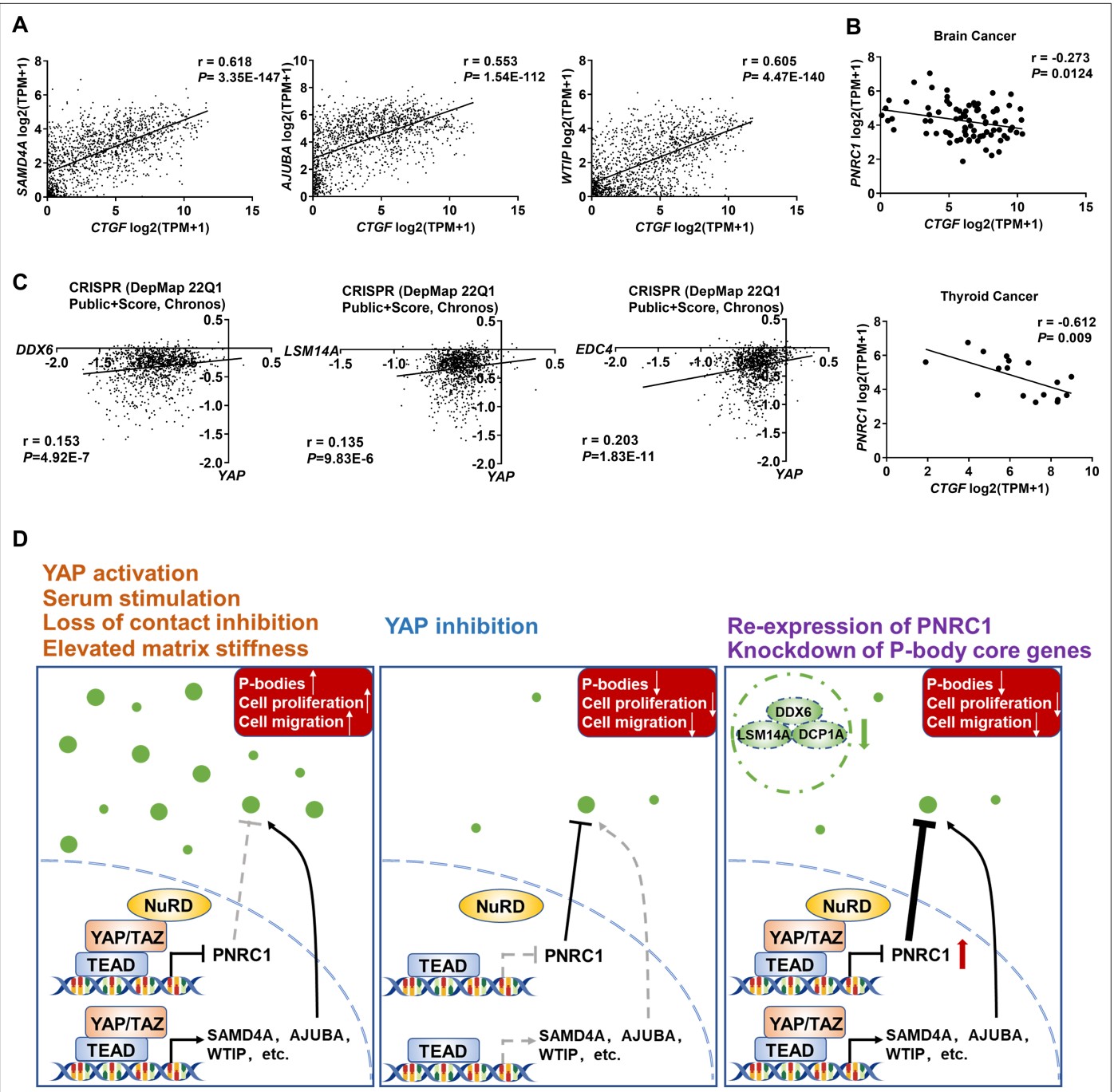

**Figure 7.** DepMap analysis reveals the co-dependencies of YAP/TEAD and P-body core genes in pancancer CRISPR screens. (**A**) Positive correlations between the mRNA levels of *CTGF* and *SAMD4A/AJUBA/WTIP* in 1393 cancer cell lines. (**B**) Negative correlation between the mRNA levels of *CTGF* and *PNRC1* in brain cancer cell lines (n = 83) and thyroid cancer cell lines (n = 17). (**C**) Positive correlations between the dependency scores of *YAP* and *DDX6/LSM14A/EDC4* in 1070 cancer cell lines. The Chronos dependency scores were extracted from the DepMap database. The negative Chronos score indicates decreased cell proliferation upon gene knockout. Pearson correlation analysis was used to assess statistical significance. (**D**) In response to serum stimulation or under loss of contact inhibition or reduced ECM stiffness, activation of YAP enhances the P-body formation to promote colorectal cancer (CRC) cell proliferation and migration. Disruption of P-bodies by overexpression of the tumor suppressor gene PNRC1 or knockdown of P-body core genes could attenuate the cell proliferation and migration induced by activation of YAP in CRC cells.

The online version of this article includes the following source data and figure supplement(s) for figure 7:

**Source data 1.** Original data for the statistical analysis in *Figure 7A–C*.

**Figure supplement 1.** Correlation analysis of the expression of YAP canonical target genes and YAP-related P-body genes.

*Figure 7 continued*

**Figure supplement 1—source data 1.** Original data for the statistical analysis in *Figure 7—figure supplement 1A–D*.

**Figure supplement 2.** The *TEAD1/3* dependency scores are positively correlated with the *EDC4/DDX6/LSM14A* scores.

**Figure supplement 2—source data 1.** Original data for the statistical analysis in *Figure 7—figure supplement 2A*.

**Figure supplement 3.** Knockdown of *DCP1A/LSM14A* and overexpression of PNRC1 suppress both cell proliferation and cell migration in A549 lung cancer cells.

**Figure supplement 3—source data 1.** Original data for the statistical analysis in *Figure 7—figure supplement 3A–C*.

**Figure supplement 3—source data 2.** Original file for the western blot analysis in *Figure 7—figure supplement 3A*.

**Figure supplement 4.** Knockdown of *DCP1A/LSM14A* and overexpression of PNRC1 suppress both cell proliferation and cell migration in MCF7 breast cancer cells.

**Figure supplement 4—source data 1.** Original data for the statistical analysis in *Figure 7—figure supplement 4A–C*.

**Figure supplement 4—source data 2.** Original file for the western blot analysis in *Figure 7—figure supplement 4A*.

**Figure supplement 5.** Knockdown of *DCP1A/LSM14A* and overexpression of PNRC1 suppress both cell proliferation and cell migration in MDA-MB-231 breast cancer cells.

**Figure supplement 5—source data 1.** Original data for the statistical analysis in *Figure 7—figure supplement 5A–C*.

**Figure supplement 5—source data 2.** Original file for the western blot analysis in *Figure 7—figure supplement 5A*.

*Castle et al., 2021*). However, an increasing number of studies have reported the contradictory role of YAP in autophagy regulation, which suggests that YAP-mediated autophagy regulation is cell type- and context-dependent (*Jin et al., 2021*; *Pei et al., 2022*; *Totaro et al., 2019*; *Wang et al., 2020*). Furthermore, though YAP is required for the cell proliferation in HUVEC, transformed cell lines often display elevated baseline YAP/TAZ activity compared to normal cells and possess many alterations in growth signaling pathways including autophagy signaling (*Nguyen and Yi, 2019*; *Shen and Stanger, 2015*; *Zanconato et al., 2016*). Thus, the contradictory observations regarding the role of YAP in modulating P-body formation between Elizabeth L. Castle et al.'s study and our study could be due to the different cell contexts and different cell conditions (baseline vs. KHSV infection).

In addition to the transcriptional regulation, P-body dynamics can also be modulated by post-translational modifications (*Luo et al., 2018*). P-body constituent proteins, such as DCP1A and DCP2, are phosphorylated, which affects the protein interaction between DCP1A and DCP2 and subsequent P-body assembly (*Chiang et al., 2013*; *Yoon et al., 2010*). Previous studies have shown that DCP1A is hyperphosphorylated during mitosis and P-body assembly is dynamically changed across the cell cycle (*Aizer et al., 2013*; *Yang et al., 2004*). Meanwhile, Hippo signaling is intrinsically regulated and YAP can also be directly phosphorylated by CDK1 during cell cycle progression (*Kim et al., 2019*; *Yang et al., 2013*). Besides, as a direct regulator of P-body formation, AJUBA is also phosphorylated by CDK1 and mitotic phosphorylation of AJUBA promotes cancer cell proliferation (*Chen et al., 2016*). Thus, we speculated that mitotic phosphorylation of YAP and AJUBA might also play a potential role in modulating P-body dynamics during cell cycle.

Compared with the role of SGs, the role of P-bodies in tumorigenesis and tumor progression is not well studied and is considered to be cancer type- or context-dependent (*Lavalée et al., 2021*). TGF-β induces P-body formation and EMT in mammary epithelial cells, while inhibition of P-body formation by knockdown of DDX6 reverses EMT and suppresses breast cancer metastasis, implying a prometastatic function of P-bodies during the progression of breast cancer (*Hardy et al., 2017*). In prostate cancer cells, dephosphorylation of EDC3 promotes the localization of EDC3-containing P-bodies and increases the P-body number (*Bearss et al., 2021*). The increase in EDC3-containing P-bodies leads to sequestration or decay of a subset of mRNAs related to cell attachment and cell growth, such as *ITGB1*, *ITGA6*, and *KLF4*, which ultimately inhibits cell proliferation and cell migration (*Bearss et al., 2021*). Of note, EDC3 and LSM14A compete for binding to the P-body core protein DDX6, and P-body formation still occurs constitutively in *EDC3* KO prostate cancer cells (*Bearss et al., 2021*). Therefore, it can be speculated that there might be different types of P-bodies that contain different RNAs and exert protumorigenic or tumor-suppressive functions in different cell contexts. In CRC, DCP1A expression is elevated, which is associated with advanced TNM stages, lymph node metastasis and poor prognosis, and overexpression of DCP1A enhances P-body formation (*Wu et al., 2018a*; *Wu et al., 2018b*). These studies imply the potential protumorigenic function of P-bodies in CRC. Furthermore,

our study showed that disruption or attenuation of P-body formation by knockdown of YAP-regulated P-body-related genes or the P-body core genes (*DDX6, DCP1A, LSM14A*) suppressed YAP-induced oncogenic phenotypes in CRC cells, such as cell proliferation and cell migration, further indicating the protumorigenic function of P-bodies in CRC or at least in CRC with active YAP. Numerous studies have demonstrated the YAP/TAZ promotes cancer cell growth through direct transcriptional regulation of genes related to cell cycle and cell apoptosis (*He et al., 2020*; *Jang et al., 2017*). Since P-bodies control the storage of untranslated mRNAs, YAP/TAZ might modulate gene expression by indirectly promoting P-body formation and the storage of untranslated target mRNAs. Future work is needed to explore P-body-enriched RNAs in CRC cells, which will further uncover the underlying mechanism by which P-bodies mediate the oncogenic function of YAP.

Recently, a study exploring new TSGs based on hemizygous deletions in multiple cancers revealed that PNRC1 is a novel tumor suppressor gene (*Gaviraghi et al., 2018*). PNRC1 translocates the cytoplasmic DCP1A/DCP2 decapping complex into the nucleolus, which subsequently impedes rRNA transcription and ribosome biogenesis (*Gaviraghi et al., 2018*). This translocation of DCP1A/DCP2 also leads to disassembly of P-bodies; thus, PNRC1 could also inhibit tumor cell proliferation by disrupting P-body formation. Moreover, hemizygous deletion of the 6q15 locus, where *PNRC1* is located, occurs in multiple cancers, including prostate, pancreatic, breast, and liver cancers (*Gaviraghi et al., 2018*). Our findings suggest that transcriptional downregulation of *PNRC1* by YAP activation could be a new mechanism of PNRC1 dysregulation during tumorigenesis. In addition, multiple oncogenes, such as MYC, RAS, and PI3K, can activate rRNA transcription and boost ribosome biogenesis to support cancer cell proliferation (*Pelletier et al., 2018*). The presence of the YAP-PNRC1 regulatory axis implies a potential role of YAP in ribosome biogenesis, which warrants further investigation in follow-up studies.

It has been shown that PNRC1 inhibits RAS- and MYC-driven tumor cell proliferation (*Gaviraghi et al., 2018*). In addition, YAP acts downstream of mutant KRAS, and activation of YAP was found to drive KRAS-independent tumor relapse in preclinical models of pancreatic cancer (*Kapoor et al., 2014*; *Shao et al., 2014*; *Zhang et al., 2014*). Strikingly, reexpression of PNRC1 also dramatically diminished the cell proliferation induced by YAP overexpression in CRC cells in our study. These data indicate that *PNRC1* is a tumor suppressor gene with a strong antitumor effect on various oncogenes; thus, reexpression of PNRC1 could be a promising anticancer therapeutic strategy. In addition, as an alternative to targeting the YAP/TEAD complex, drugs that inhibit downstream effectors of YAP/TAZ have shown efficacy in the clinic (*Gay et al., 2017*; *Neesse et al., 2013*; *Nguyen and Yi, 2019*). The identification of the P-body as a new downstream effector of YAP/TAZ suggests that disruption of P-bodies might be a potential therapeutic strategy for tumors with active YAP. Although each P-body core gene performs multiple biological functions, unbiased functional CRISPR screening across cancer cell lines (DepMap) revealed that loss of function of a series of P-body core genes significantly suppresses proliferation in various tumor cell lines. The functional overlap in P-body assembly and the positive correlation between the dependency profiles of these P-body core genes imply the important role of P-bodies in tumor cell proliferation and cell survival. Several compounds, including translation inhibitors, have been reported to inhibit P-body formation (*Martínez et al., 2013*; *Stribinskis and Ramos, 2007*). Notably, actin polymerization can activate YAP (*Sun and Irvine, 2016*). Methylchivosazol, an actin polymerization inhibitor, was found to be a strong inhibitor of P-body formation by screening of a library of compounds derived from myxobacteria (*Martínez et al., 2013*). However, these small molecules indirectly target P-bodies and show extensive effects on cells (*Martínez et al., 2013*; *Stribinskis and Ramos, 2007*). Thus, the development of inhibitors directly targeting P-body core proteins will provide a chemical tool for exploring the function of P-bodies in tumors and assess the therapeutic efficacy of P-body disassembly in cancer. Overall, our study reveals the P-body as a new downstream effector of YAP/TAZ, which opens a new possibility of targeting P-body assembly to combat tumors (*Figure 7D*).

## Materials and methods
### Cell culture and transfection
HEK293T, NIH3T3, HCT116, MCF7, MDA-MB-231, and A549 cells were purchased from the American Type Culture Collection (ATCC) and authenticated by short tandem repeat analysis. HEK293T,

NIH3T3, HCT116, and MDA-MB-231 cells were cultured in Dulbecco's Modified Eagle Medium (DMEM)/high-glucose (HyClone) supplemented with 10% fetal bovine serum (FBS), 100 units/ml penicillin, and 100 µg/ml streptomycin (Sangon Biotech) at 37°C in 5% $CO_2$. MCF7 cells were cultured in Minimum Essential Medium (MEM)/high-glucose (HyClone), A549 cells were cultured in RPMI-1640 high-glucose medium (HyClone), and the other culture conditions were the same as those used for HCT116 cells. Corning TC-treated Culture Dish was used for routine cell culture. For the cell culture with 2D polyacrylamide-based hydrogels, hydrogels of high (40.40 ± 2.39 kPa) or low (1.00 ± 0.31 kPa) stiffness were generated according to the published protocol (*Tse and Engler, 2010*). Fibronectin solution (3 µg/ml) and Sigmacote were used to coat the surface of the hydrogels. PEI (Polysciences) and Lipofectamine 2000 (Invitrogen) were used for plasmid transfection. Lipofectamine RNAiMAX (Invitrogen) was used for siRNA transfection.

## Plasmids and reagents

Full-length *PNRC1* cDNA was inserted into the pQCXIH vector, and the mutant pQCXIH-FLAG-*PNRC1*^W300A^ plasmid was constructed by using a KOD mutagenesis kit (Toyobo, Osaka, Japan) according to the manufacturer's instructions. The pBABE-FLAG-*YAP*$^{5SA}$/*YAP*$^{5SA-S94A}$ and pRK7-FLAG-*YAP*$^{5SA}$/*YAP*$^{5SA-S94A}$ plasmids were obtained from laboratory storage. To generate the shRNA constructs targeting human *TAZ*, the targeting sequences were inserted into the pLKO.1-puro vector. The shRNA constructs targeting human *YAP, SAMD4A, AJUBA, DDX6, DCP1A,* and *LSM14A* were generated by using the pLKO.1-hygro vector. The *PNRC1* promoter and intron reporter plasmids and TEAD binding site mutant reporter plasmids were constructed by using the pGL3-Basic vector. siRNA/shRNA was used for all RNA silencing experiments in the study. The siRNA oligos targeting *CHD4, RBBP4, PNRC1, SAMD4A, AJUBA, YAP,* and *TAZ* were synthesized by Shanghai Genepharma Co., Ltd. The shRNA and siRNA targeting sequences and the primers used for plasmid construction are listed in *Supplementary file 3*. The following antibodies were used in this study: anti-FLAG (D6W5B, CST), anti-FLAG (M2, Sigma), anti-YAP/TAZ (D24E4, CST), anti-YAP (sc-101199, Santa Cruz), anti-TEAD4 (ab58310, Abcam), anti-CHD4 (14173-1-AP, Proteintech), anti-DDX6 (A9634, ABclonal), anti-DCP1A (A6824, ABclonal), anti-LSM14A (18336-1-AP, Proteintech), anti-AJUBA (A22039, ABclonal), anti-SAMD4A (17387-1-AP, Proteintech), and anti-PNRC1 (51052-1-AP, Proteintech).

## Cell proliferation, colony formation, and cell migration assays

For the proliferation assay, cells (1 × 10³ per well) were seeded into a 96-well plate and cultured for 5 d. Cell viability was measured every day with a Cell Counting Kit 8 (CCK8) (Vazyme) according to the manufacturer's instructions. Briefly, 10 µl of CCK8 reagent was added to each well and incubated for 2 hr. The absorbance at 450 nm was measured with a microplate reader to determine the relative numbers of viable cells. For the colony formation assay, cells (1 × 10³ per well) were seeded and cultured in six-well plates for 2 wk. Then, the cells were stained with 1% crystal violet, and the number of colonies in each well was counted. For the cell migration assay, cells (1.5 × 10⁵ per well) in DMEM/high-glucose containing 0.1% FBS were seeded in the upper compartment of a Transwell chamber, while DMEM/high-glucose containing 10% FBS was placed in the lower compartment. After 60 hr, migrated cells were stained with 1% crystal violet and counted.

## Cell cycle and cell apoptosis assays

Cell cycle and cell apoptosis assays were performed according to the manufacturer's instruction (FITC Annexin V Apoptosis Detection Kit I, BD, 556547; PI/RNase staining buffer, BD, 550825).

## qPCR, ChIP, and luciferase reporter assays

qPCR and ChIP were performed as previously described (*Zhu et al., 2020*). For the luciferase assay, HEK293T cells were seeded in 24-well plates, incubated overnight to 50% confluence, and then co-transfected with the *PNRC1* luciferase reporter plasmid and the FLAG-*YAP*$^{5SA}$ or FLAG-*YAP*$^{5SA-S94A}$ plasmid. A Renilla luciferase plasmid was used as the control. After 24–36 hr, luciferase activity was measured by using a dual-luciferase reporter assay (Promega) and normalized to Renilla luciferase activity. For the luciferase assay using the *YAP/TAZ* knockdown of HCT116 cells, stable HCT116 cells were seeded in 24-well plates and co-transfected with the *PNRC1* luciferase reporter and Renilla

luciferase plasmids for 6 hr. Then, cells were replated to six-well plates and cultured for 24–36 hr at a low cell density before measuring the luciferase activity.

### Immunofluorescence staining

Cells were seeded in glass-bottom cell culture dishes one night before IF staining. The cells were washed with phosphate-buffered saline (PBS) and fixed with 4% paraformaldehyde. After the cells were permeabilized with 0.1% Triton X-100 at room temperature (RT) for 5 min, they were blocked with 3% Bovine Serum Albumin (BSA) at RT for 1 hr. Then, the cells were incubated overnight with a rabbit anti-DDX6/DCP1A/LSM14A antibody (1:250) or a mouse anti-FLAG (M2) antibody (1:150) or mouse anti-YAP antibody (1:100). After a 1 hr incubation with Cy3-conjugated mouse and Alexa Fluor 488-conjugated rabbit secondary antibodies followed by a 1 min incubation with 0.1 µg/ml DAPI, the cells were visualized with an Olympus IX81 microscope.

### Xenograft assay and immunohistochemistry

Nude mice (4–6 weeks old, male) were obtained from SLAC Laboratory Animals LLC, Shanghai, China. All mouse procedures were approved by the Xinhua Hospital Animal Care and Use Committee. Male nude mice (4–6 weeks old) were randomly divided into four groups (n = 6 mice per group) and injected in the right flank with $2 \times 10^6$ of the indicated stable HCT116 cells resuspended in 100 µl of PBS. Mice were sacrificed on day 21, and the xenograft tumors were removed, photographed, and paraffin embedded for sectioning. The xenograft tumors were sectioned for H&E staining and immunohistochemical staining with anti-ki67 and anti-PNRC1 antibodies as described previously (*Zhu et al., 2020*).

### Colorectal cancer specimen

Patients with CRC who underwent curative surgery without prior treatments at the Department of Colorectal and Anal Surgery, Xinhua Hospital, Shanghai Jiao Tong University School of Medicine, between January 2008 and December 2018 were enrolled. Institutional review board approval and informed consent were obtained for all sample collections. None of the patients had any history of other tumors. Tumors and paired paracancerous normal tissues were collected during surgery. The generation of the CRC tissue array has been described, and the IHC analysis of the CRC tissue array was performed according to our previous study (*Zhu et al., 2020*).

### Statistical analysis

Statistical analysis was performed by using GraphPad Prism 8.0.2 and SPSS 22.0. Typically, differences between two groups were evaluated using two-tailed Student's *t*-test or the chi-square test as indicated in the figure legends. One-way ANOVA was used for the experiments with more than two groups. Mann–Whitney *U* test and Kruskal–Wallis test were used for statistical analysis of the foci number in IF staining assay. Two-way ANOVA was used for statistical analysis of CCK8 assay. Paired Student's *t*-test was performed to assess the statistical significance of differential *PNRC1* mRNA expression in 16 pairs of CRC and adjacent normal tissues. Pearson correlation analysis was used to statistically assess correlations of mRNA levels or co-dependencies from the DepMap database. The results are shown as averages; the error bars indicate the SDs. p-Values less than 0.05 were considered significant (*p<0.05; **p<0.01; ***p<0.001; ****p<0.0001).

## Acknowledgements

This work was sponsored by the National Key R&D Program of China (2019YFC1316002), the National Natural Science Foundation of China (82172916, 82372648, 82073056, 82273022), and the Program for Professor of Special Appointment (Eastern Scholar) at Shanghai Institutions of Higher Learning (to C-YL).

# Additional information

## Funding

| Funder | Grant reference number | Author |
| --- | --- | --- |
| National Natural Science Foundation of China | 82172916 | Chen-Ying Liu |
| National Natural Science Foundation of China | 82073056 | Yun Liu |
| National Natural Science Foundation of China | 82273022 | Long Cui |
| Program for Professor of Special Appointment (Eastern Scholar) at Shanghai Institutions of Higher Learning | | Chen-Ying Liu |
| National Natural Science Foundation of China | 82372648 | Chen-Ying Liu |

The funders had no role in study design, data collection and interpretation, or the decision to submit the work for publication.

## Author contributions

Xia Shen, Xiang Peng, YueGui Guo, Investigation, Methodology; Zhujiang Dai, Software; Long Cui, Resources, Funding acquisition; Wei Yu, Validation; Yun Liu, Data curation, Supervision, Funding acquisition; Chen-Ying Liu, Conceptualization, Supervision, Funding acquisition, Writing – original draft, Writing – review and editing

## Author ORCIDs

Wei Yu http://orcid.org/0000-0001-9898-4607
Yun Liu http://orcid.org/0000-0002-0258-4025
Chen-Ying Liu https://orcid.org/0000-0002-8930-6182

## Ethics

Patients with colorectal cancer who underwent curative surgery without prior treatments at the Department of Colorectal and Anal Surgery, Xinhua Hospital, Shanghai Jiao Tong University School of Medicine, between January 2008 and December 2018 were enrolled.

All mouse studies were approved, and all animals were manipulated according to the protocols approved by the Animal Care and Use Committees of Xinhua Hospital and animal care was conducted in accordance with institutional guidelines. According to the criteria of the Animal Care and Use Committee of Xinhua Hospital, the maximal tumor burden permitted was <10% body weight, at no point did any mice exceed maximal tumor burden. Mice were housed in pathogen-free and ventilated cages, and allowed free access to irradiated food and autoclaved water ad libitum in a 12 h light/dark cycle, with room temperature at 21 °C and humidity between 45 and 65%. Male BALB/c nude mice of ~4-6 weeks of age were purchased from GemPharmatech, Shanghai, China.

Reviewer #2 (Public Review): https://doi.org/10.7554/eLife.88573.3.sa1
Author response https://doi.org/10.7554/eLife.88573.3.sa2

# Additional files

## Supplementary files

• Supplementary file 1. The expression of P-body genes in YAP/TAZ knockdown HCT116 cells. The differentially expressed genes in YAP/TAZ knockdown HCT116 cells were analyzed by RNA-seq (GSE176475). The expression levels of genes that were annotated as related to P-bodies are included in this supplementary file.

• Supplementary file 2. Top 100 correlation for EDC4 in CRISPR (DepMap 22Q1 Public Score,

Chronos). The gene dependency scores of the top 100 genes that are correlated with EDC4 were downloaded from the Cancer Dependency Map (DepMap) database.

• Supplementary file 3. siRNA, shRNA, and primers used in this study. The sequences of the siRNA, shRNA, and primers used in this study are included in this supplementary file.

• MDAR checklist

## Data availability

All WB source data generated during this study are included in the manuscript and supporting files. Source data files have been provided for figures.

The following previously published dataset was used:

| Author(s) | Year | Dataset title | Dataset URL | Database and Identifier |
|---|---|---|---|---|
| Guo Y, Zhu Z, Liu C | 2022 | Identification of differential expressed genes after HHEX/YAP/TAZ/TEAD knockdown in colorectal cancer cells | https://www.ncbi.nlm.nih.gov/geo/query/acc.cgi?acc=GSE176475 | NCBI Gene Expression Omnibus, GSE176475 |

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
