## [Editor Report · eLife assessment]

This **valuable** study advances our understanding that YAP/TAZ, as well as their target genes, plays a prominent role in the formation of processing bodies (P-bodies). The evidence supporting the conclusions is **convincing**. The article could be improved through further analysis to elucidate the mechanistic link between P-body formation and oncogenesis. The work will be of broad interest to scientists working in the field of Hippo signaling and cancer biology.

---

## [Referee Report · Reviewer #2 (Public Review)]

In a study by Shen et al.. al., the authors investigated YAP/TAZ target genes that play a role in the formation of processing bodies (P-bodies). P-bodies are membraneless cytoplasmic granules that contain translationally repressed mRNAs and components of mRNA turnover. GO enrichment analysis of the RNA-Seq data of colorectal cancer cells (HCT116) after YAP/TAZ knockdown showed that the downregulated genes were enriched in P-body resident proteins. Overexpression, knockdown, and ChIP-qPCR analyses showed that SAMD4A, PNRC1, AJUBA, and WTIP are YAP-TEAD target genes that also play a role in P-body biogenesis. Using P-body markers such as DDX6 and DCP1A, the authors showed that knockdown of YAP in the HCT116 cell line causes a reduction in the number of P-bodies. Similarly, overexpression of constitutively active YAP (YAP 5SA) increased the P-body number. The YAP-TEAD target genes SAMD4A and AJUBA positively regulate P-body formation, because lowering their expression levels using siRNA reduces the number of P-bodies. The other YAP target gene, PNRC1, is a negative regulator of P-body biogenesis and consistently YAP suppresses its expression through the recruitment of the NuRD complex. YAP target genes that modulate P-body formation play prominent roles in oncogenesis. PNRC1 suppression is key to YAP-mediated proliferation, colony formation, and tumorigenesis in HCT116 xenografts. Similarly, SAMD4 and AJUBA knockdown abrogated cell viability. In summary, this study demonstrated that SAMD4, AJUBA, WTIP, and PNRC1 are bona fide YAP-TEAD target genes that play a role in P-body formation, which is also linked to the oncogenesis of colon cancer cells.

Major Strengths:

The majority of the experiments were appropriately planned so that the generated data could support the conclusions drawn by the authors. The phenotype observed with YAP/TAZ knockdown correlated inversely with YAP5SA overexpression, which is complementary. Where possible, the authors also used point mutations that selectively disrupt protein-protein interactions, such as YAP S94A and PNRC1 W300A. The CRC cell line HCT116 was used throughout the study; additionally, data from other cancer cell lines were used to support the generality of the findings.

Weaknesses:

The authors did not elucidate the mechanistic link between P-body formation and oncogenesis; therefore, it is unclear why an increase in the number of P-bodies is pro-tumorigenic. The authors extrapolated and suggested that PNRC1 expression could be exploited therapeutically, without providing much detail. How do they plan to stimulate the expression of PNRC1? It is not necessary for every scientific finding to lead to a therapeutic benefit; therefore, they can tone down such statements if therapeutic exploitation is not realistic. The authors elucidated a mechanism for PNRC1 repression and one wonders why no attempts were made to understand the mechanism of activation of SAMD4, AJUBA, and WTIP expression.

---

## [Author Response]

The following is the authors’ response to the original reviews.

**Reviewer #1 (Public Review):**
In this manuscript, the authors demonstrated that YAP/TAZ promotes P-body formation in a series of cancer cell lines. YAP/TAZ modulates the transcription of multiple P-body-related genes, especially repressing the transcription of the tumor suppressor proline-rich nuclear receptor coactivator 1 (PNRC1) through cooperation with the NuRD complex. PNRC1 functions as a critical repressor in YAP-induced biogenesis of P-bodies and tumorigenesis in colorectal cancer (CRC). Reexpression of PNRC1 or disruption of P-bodies attenuated the protumorigenic effects of YAP. Overall, these findings are interesting and the study was well conducted.

We thank the reviewer for the positive comments for our work.

Major concerns:(1) RNAseq data indicated that Yap has the capacity to suppress the expression of numerous genes. In addition to PNRC1, could there be additional Yap targeting factors involved in Yap-mediated the formation of P-bodies?

Yes, indeed. Additional YAP target genes, such as AJUBA, SAMD4A, are also involved in YAP-mediated the formation of P-bodies (Fig. 1B-D). Knockdown of either SMAD4A or AJUBA attenuated the P-body formation induced by overexpression of YAP5SA (Fig. 3A).

(2) It is still not clear how PNRC1 regulates P-bodies. Knockdown of PNRC1 prevented the reduction of P-bodies caused by Yap knockdown. How do the genes related to P-bodies that are positively regulated by Yap, such as SAMD4A, AJUBA, and WTIP, change in this scenario? Given that the expression of Yap can differ considerably among various cell types, is it possible for P-bodies to be present in tumor cells lacking Yap expression?

The detail mechanism of PNRC1’s suppressive effect on P-body formation was well explored in Gaviraghi et al.’s paper, in which PNRC1 was first identified as a tumor suppressor gene (EMBO, 2018, PMID: 30373810). Gaviraghi et al. revealed that overexpression of PNRC1 leads to translocation of cytoplasmic DCP1A/DCP2 into the nucleolus, which subsequently attenuates rRNA transcription and ribosome biogenesis. Since DCP1A and DCP2 are essential for formation of P-bodies, loss of cytoplasmic DCP1A/DCP2 also disrupts P-body formation. This background information has been included in the Results and Discussion sections in the manuscript:

Previously, we have performed the RNA-seq analysis of HCT116 cells with overexpression of PNRC1. Compared with YAP5SA overexpression (520 differentially expressed genes), overexpression of PNRC1 showed less effect on the gene expression profile (147 differentially expressed genes) and expression of SAMD4A, AJUBA and WTIP were not affected by PNRC1 overexpression.

In this study, we found that YAP could promote P-body formation in a series of cancer cell lines. During the exploration, we observed that P-bodies hardly existed in the RKO colorectal cancer cell line (Figure 1 for the reviewer). However, the regulatory effect of YAP/TAZ on SAMD4A, AJUBA, and WTIP was still observed (Figure 2 for the reviewer). These data suggest that YAP’s activity could be sufficient but not required for the P-body formation. So, we agree that P-bodies could be present in tumor cells lacking Yap expression.

**Author response image 1. sa2fig1:** 

(3) The authors demonstrated that CHD4 can bind to Yap target genes, such as CTGF, AJUBA, SAMD4A (Figure 4 - Figure Supplement 1D). Does the NuRD complex repress the expression of these genes? the NuRD complex could prevent the formation of P-bodies?

Good point! Following the reviewer’s suggestions, we detected the mRNA levels of AJUBA, WTIP and SAMD4A, and the P-body formation the CHD4 knockdown cells. Interestingly, knockdown of CHD4 induced mild downregulation of AJUBA, WTIP and SAMD4A in HCT116 cells (Figure 3 for the reviewer). Of note, NuRD complex is involved in both transcriptional repression and activation (PNAS 2011, PMID: 21490301; Stem Cell Reports. 2021, PMID: 33961790). As expected, knockdown CHD4 induced decreased number of P-bodies in HCT116 cell (new Figure 4-Supplement 1E), which is consistent to the enhanced expression of PNRC1 (Figure 4F).

**Author response image 3. sa2fig3:** 

**Author response image 4. sa2fig4:** 

(4) YAP/TAZ promotes the formation of P-bodies which contradicts the previous study's conclusion (PMID: 34516278). Please address these inconsistent findings.

The contradictory observations between our and the previous studies could be due to the different cell lines (HUVEC vs cancer cell lines) and different stimuli (KHSV infection vs normal culture condition or serum stimulation, cell density and stiffness). Actually, we have discussed the contradictory observation in the previous study in the Discussion section as followed:

“In contrast, a recent study, which provided the first link between YAP and P-bodies, implicated YAP as a negative regulator of P-bodies in KHSV-infected HUVECs (Castle et al, 2021). Elizabeth L. Castle et al. reported that virus-encoded Kaposin B (KapB) induces actin stress fiber formation and disassembly of P-bodies, which requires RhoA activity and the YAP transcriptional program (Castle et al, 2021). YAP-enhanced autophagic flux was proposed to participate in KapB-induced P-body disassembly, consistent with the concept that stress granules and P-bodies are cleared by autophagy (Buchan et al, 2013; Castle et al, 2021). However, an increasing number of studies have reported the contradictory role of YAP in autophagy regulation, which suggests that YAP-mediated autophagy regulation is cell type- and context-dependent (Jin et al, 2021; Pei et al, 2022; Totaro et al, 2019; Wang et al, 2020). Furthermore, though YAP is required for the cell proliferation in HUVEC, transformed cell lines often display elevated baseline YAP/TAZ activity compared to normal cells and possess many alterations in growth signaling pathways including autophagy signaling (Nguyen & Yi, 2019; Shen & Stanger, 2015; Zanconato et al, 2016). Thus, the contradictory observations regarding the role of YAP in modulating P-body formation between Elizabeth L. Castle et al.’s study and our study could be due to the different cell contexts and different cell conditions (baseline vs. KHSV infection).”

**Reviewer #2 (Public Review):**
In a study by Shen et al., the authors investigated YAP/TAZ target genes that play a role in the formation of processing bodies (P-bodies). P-bodies are membraneless cytoplasmic granules that contain translationally repressed mRNAs and components of mRNA turnover. GO enrichment analysis of the RNA-Seq data of colorectal cancer cells (HCT116) after YAP/TAZ knockdown showed that the downregulated genes were enriched in P-body resident proteins. Overexpression, knockdown, and ChIP-qPCR analyses showed that SAMD4A, PNRC1, AJUBA, and WTIP are YAP-TEAD target genes that also play a role in P-body biogenesis. Using P-body markers such as DDX6 and DCP1A, the authors showed that the knockdown of YAP in the HCT116 cell line causes a reduction in the number of P-bodies. Similarly, overexpression of constitutively active YAP (YAP 5SA) increased the P-body number. The YAP-TEAD target genes SAMD4A and AJUBA positively regulate P-body formation, because lowering their expression levels using siRNA reduces the number of P-bodies. The other YAP target gene, PNRC1, is a negative regulator of P-body biogenesis and consistently YAP suppresses its expression through the recruitment of the NuRD complex. YAP target genes that modulate P-body formation play prominent roles in oncogenesis. PNRC1 suppression is key to YAP-mediated proliferation, colony formation, and tumorigenesis in HCT116 xenografts. Similarly, SAMD4 and AJUBA knockdown abrogated cell viability. In summary, this study demonstrated that SAMD4, AJUBA, WTIP, and PNRC1 are bona fide YAP-TEAD target genes that play a role in P-body formation, which is also linked to the oncogenesis of colon cancer cells.

We thank the reviewer for the positive comments for our work.

Major Strengths:The majority of the experiments were appropriately planned so that the generated data could support the conclusions drawn by the authors. The phenotype observed with YAP/TAZ knockdown correlated inversely with YAP5SA overexpression, which is complementary. Where possible, the authors also used point mutations that selectively disrupt protein-protein interactions, such as YAP S94A and PNRC1 W300A. The CRC cell line HCT116 was used throughout the study; additionally, data from other cancer cell lines were used to support the generality of the findings.

We thank the reviewer for the positive comments regarding the strength and significance of our work.

Weaknesses:The authors did not elucidate the mechanistic link between P-body formation and oncogenesis; therefore, it is unclear why an increase in the number of P-bodies is pro-tumorigenic. AJUBA and SAMD4 may have housekeeping functions and reduce the proliferation of YAP-independent cell lines. Figure 6 - Figure Supplement 4 shows a reduction in cell viability and migration in control HCT116 cell lines upon AJUBA/SAMD4 knockdown. Therefore, it is unclear whether their tumor suppressive role is YAP-dependent. The authors extrapolated and suggested that their findings could be exploited therapeutically, without providing much detail. How do they plan to stimulate the expression of PNRC1? It is not necessary for every scientific finding to lead to a therapeutic benefit; therefore, they can tone down such statements if therapeutic exploitation is not realistic. The authors elucidated a mechanism for PNRC1 repression and one wonders why no attempts were made to understand the mechanism of activation of SAMD4, AJUBA, and WTIP expression.

We thank the reviewer for pointing out these issues to further improve the quality of our study. As mentioned in the Abstract section, the role of P-bodies in tumorigenesis and tumor progression is not well studied. In this study, we revealed that disruption of P-body formation by knockdown of essential P-body-related genes attenuates YAP-driven oncogenic function in CRC, which provides evidence implicating the pro-tumorigenic role of P-bodies. We agree with the reviewer that the mechanism of P-body formation promoting tumorigenesis is an important scientific question warranting exploration and plan to investigate this fancy question in next study.

AJUBA has been known to act as a signal transducer in oncogenesis and promote CRC cell survival (Pharmacol Res. 2020, PMID: 31740385; Oncogene. 2017, PMID: 27893714). Furthermore, as the reviewer suggested, we found that knockdown of both AJUBA and SAMD4A suppressed the cell proliferation in the YAP-deficient cell line, SHP-77, which further implicates the oncogenic role of AJUBA and SAMD4A (Figure 4 for the reviewer). Numerous studies have shown that YAP/TAZ knockdown suppressed the cell proliferation of HCT116 cells. Thus, not surprisingly, knockdown of AJUBA and SAMD4A also repressed the cell proliferation of the “parental” control HCT116 cells. Since the molecular mechanistic studies identified the AJUBA and SAMD4A were bona fide YAP-TEAD target genes, the co-dependencies of YAP and AJUBA/SAMD4A in the HCT116 cells imply that the pro-tumorigenic function of YAP could be dependent on activation of AJUBA/SAMD4A, in some extent (due to the large amount of YAP target genes).

**Author response image 5. sa2fig5:** 

Tumor suppressor genes are frequently epigenetically silenced in cancer cells, so is PNRC1. In our preliminary study, we found that the DNA methyltransferase inhibitor 5-Azacytidine dramatically increased the mRNA level of PNRC1 in HCT116 cells (Figure 5 for the reviewer), which suggests that PNRC1 is epigenetically suppressed by DNA methylation in CRC cells and could be re-activated or re-expressed by DNA methyltransferase inhibitor for the cancer treatment.

**Author response image 6. sa2fig6:** 

YAP/TAZ are well-known as transcriptional co-activators and the mechanism of transcriptional activation of target genes has been well-studied (Cell Stress. 2021, PMID: 34782888). However, years later, the function of YAP/TAZ as the transcriptional co-repressors was brought to the forefront. Both NuRD and Polycomb repressive complex 2 (PRC2) are involved in the transcriptional repressor function of YAP (Cell Rep. 2015, PMID: 25843714; Cancer Res. 2020, PMID: 32409309). Thus, we focused on exploring mechanism for PNRC1 repression in this study, but not the mechanism of activation of SAMD4A, AJUBA, and WTIP expression.

**Reviewer #2 (Recommendations For The Authors):**
Suggested experiments: The suggested experiments were aimed at minimizing the weaknesses of the manuscript. The roles of AJUBA and SAMD4 can be elucidated in a YAP-independent cell line. After knockdown of AJUBA or SAMD4 in a YAP-independent cell line, the effects on proliferation and migration should be determined.

Following the reviewer’s suggestions, we explored the role of AJUBA and SAMD4A in the YAP-independent cell line, SHP-77 (Cancer Cell. 2021, PMID: 34270926). Unfortunately, SHP-77 cells are suspension cells mixed with some loosely adherent cells, and we found that SHP-77 cells are not available for cell migration assay. By CCK8 assay, we found that knockdown of both AJUBA and SAMD4A suppressed the cell proliferation in SHP-77 cells, which further implicates the oncogenic role of AJUBA and SAMD4A.

**Author response image 7. sa2fig7:** 

Experiments directed at elucidating whether the mRNAs of tumor suppressor genes undergo sequestration and decay in P-bodies that ultimately promote tumorigenesis will provide a mechanistic link between P-body formation and tumorigenesis. The enrichment of P-bodies through biochemical methods has been employed in other studies. RNA-seq after P-body enrichment may provide opportunities to unravel the link between P-body formation and tumorigenesis.

We thank the reviewer for the constructive suggestions to further improve the significance of our study. We do have plans to purify the P-bodies to further elucidate underlying mechanisms of pro-tumorigenic role of P-bodies tumor cells. However, we are newcomers in the P-body field and encountered a lot of issues to establish the biochemical assays of P-bodies. Hopefully, we can solve these technical issues soon and present our new data in the next paper.